# Computational study on the catalytic control of *endo*/*exo* Diels-Alder reactions by cavity quantum vacuum fluctuations

Fabijan Pavošević [1] ✉, Robert L. Smith [1,2] & Angel Rubio [1,3,4] ✉

Achieving control over chemical reaction's rate and stereoselectivity realizes one of the Holy Grails in chemistry that can revolutionize chemical and pharmaceutical industries. Strong light-matter interaction in optical or nano-plasmonic cavities might provide the knob to reach such control. In this work, we demonstrate the catalytic and selectivity control of an optical cavity for two selected Diels-Alder cycloaddition reactions using the quantum electro-dynamics coupled cluster (QED-CC) method. Herein, we find that by changing the molecular orientation with respect to the polarization of the cavity mode the reactions can be significantly inhibited or selectively enhanced to produce major *endo* or *exo* products on demand. This work highlights the potential of utilizing quantum vacuum fluctuations of an optical cavity to modulate the rate of Diels-Alder cycloaddition reactions and to achieve stereoselectivity in a practical and non-intrusive way. We expect that the present findings will be applicable to a larger set of relevant reactions, including the click chemical reactions.

Diels-Alder cycloaddition reactions[1], in which a conjugated diene and an alkene (dienophile) form a cyclohexene derivative, are one of the most iconic chemical reactions used to form carbon-carbon bonds. Due to their unique versatility, Diels-Alder cycloaddition reactions have played an essential role in virtually all fields of synthetic chemistry, such as click chemistry[2], total synthesis of certain natural products[3], biosynthesis[4], and in the synthesis of thermoplastic polymers[5]. In addition to their unrivaled synthetic significance, they also played a crucial role in the development of the famous Woodward-Hoffmann rules[6] for chemical reactivity.

A key feature of the Diels-Alder cycloaddition reaction is the ability to form two distinct *endo* and *exo* diastereomeric products when the dienophile is a substituted alkene. The *endo* product results from a transition state structure in which a dienophile's substituent lies above the diene, whereas the *exo* product results from a transition state structure in which a dienophile's substituent is oriented away from it (Fig. 1a). Due to the fundamental and technological relevance of

the Diels-Alder cycloaddition reactions, several techniques have been proposed to control the rate and *endo*/*exo* selectivity, such as antibody[7], enzymatic[8], or electric field[9] control.

Another path to control chemical reactions by using strong light-matter coupled hybrid states (polaritonic states)[10] has been opened recently[11–26]. Strongly coupled polaritonic states can be created by either quantum fluctuations or by external pumping in optical or nanoplasmonic cavities[27]. Inside the cavity, molecular polaritons are formed by direct coupling of the molecular complex to the cavity quantum fluctuations. Because the properties of molecular polaritons can be modulated by the strength of the light matter coupling, this technique offers a non-intrusive way to catalyze[15], inhibit[12], or even change the reaction path[14] of a given chemical reaction. Most of the experimental works for controlling chemical reactions are done in the strongly coupled vibro-polariton regime[12–14,16,17], however in this work our focus is on the electronic polariton effects on the chemistry as done for exciton-polaritons in driven 2D materials in cavities[28,29].

[1]Center for Computational Quantum Physics, Flatiron Institute, 162 5th Ave., New York 10010 NY, USA. [2]Department of Chemistry, Virginia Tech, Blacksburg, VA 24061, USA. [3]Max Planck Institute for the Structure and Dynamics of Matter and Center for Free-Electron Laser Science & Department of Physics, Luruper Chaussee 149, 22761 Hamburg, Germany. [4]Nano-Bio Spectroscopy Group and European Theoretical Spectroscopy Facility (ETSF), Universidad del País Vasco (UPV/EHU), Av. Tolosa 72, 20018 San Sebastian, Spain. ✉e-mail: fpavosevic@gmail.com; angel.rubio@mpsd.mpg.de

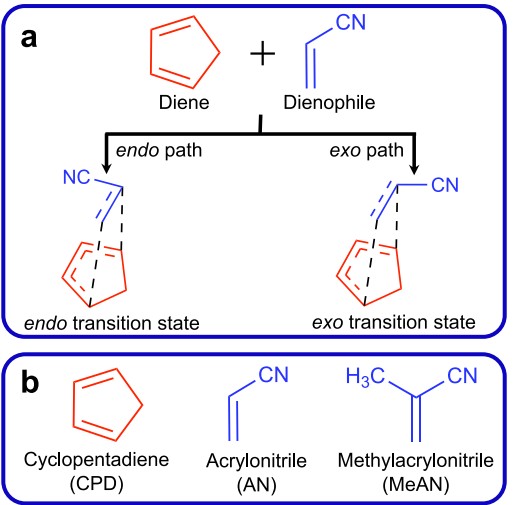

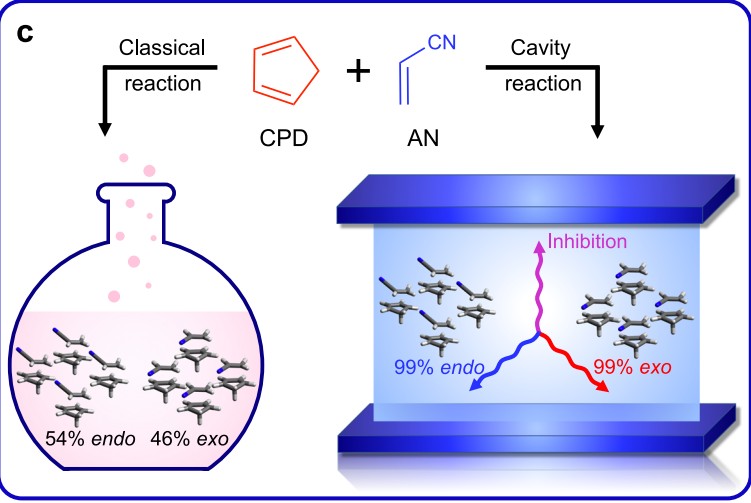

**Fig. 1 | Cavity controlled catalysis and selectivity in Diels–Alder reactions. a** The Diels-Alder cycloaddition reaction can form two different *endo* and *exo* diastereomers. These two distinct products are formed in two different paths determined by the relative orientation of the dienophile molecule. **b** Reactants for the Diels-Alder cycloaddition reactions considered in this work; cyclopentadiene (CPD), acrylonitrile (AN), and methylacrylonitrile (MeAN). **c** Schematic summary of the findings from this work. For the Diels-Alder cycloaddition between CPD and AN at room temperature, the classical reaction gives a 54:46 *endo/exo* product ratio, whereas the same reaction in a cavity gives 99% of *endo* or *exo* product, depending on the light mode polarization direction.

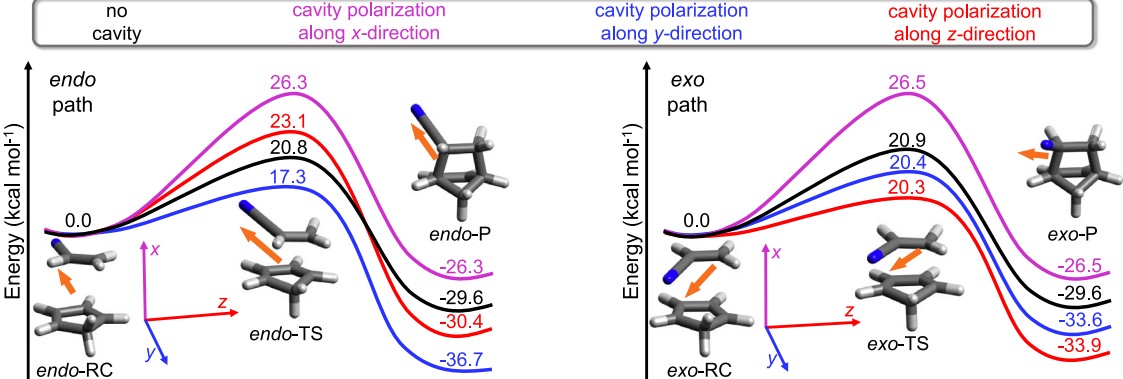

**Fig. 2 | Reaction diagram of Diels-Alder cycloaddition reaction between cyclopentadiene (CPD) and acrylonitrile (AN).** Reaction energy diagram for *endo* path (left panel) and *exo* path (right panel) calculated with the CCSD/cc-pVDZ (black) and QED-CCSD/cc-pVDZ methods. The quantum electrodynamics (QED) calculations employ the cavity frequency $\omega_{cav} = 1.5$ eV and light-matter coupling strength $|\lambda| = 0.1$ a.u. with one photon mode polarized in the $x$ (magenta), $y$ (blue), and $z$ (red) molecular directions (marked in the inset of the figure). Both panels contain the images of reaction complex (RC), transition state (TS), and product (P) structures, their dipole moment directions (orange arrow), and the molecular coordinate frame, where $x$ direction is along the forming carbon-carbon bond, and $y$ and $z$ directions lay in plane of the CPD ring. Source data are provided as a Source Data file.

Naturally, in order to understand and explain these experiments that operate in the exciton-polariton regime, ab initio methods in which electrons and photons are treated quantum mechanically on equal footing are required. Recent theoretical advances in which electronic structure methods and quantum electrodynamics are combined[20,30–34] have provided a significant insight into chemical reactions inside an optical cavity where selected reactions are either enhanced, inhibited, or steered[20,24,35–38], however a single chemical process in which one either enhances, inhibits, or steers a chemical reaction has yet to be demonstrated.

In this work, we demonstrate that the quantum vacuum fluctuations of a cavity can be used as a viable tool for selective controll of catalysis. To achieve that goal, we extend the recently developed quantum electrodynamics coupled cluster (QED-CC) method[30–33] to investigate the effect of an optical cavity on the reaction rate and *endo/exo* stereoselectivity of the Diels-Alder cycloaddition reaction. The two target Diels-Alder cycloaddition reactions considered herein are reactions of cyclopentadiene (CPD) with acrylonitrile (AN) and

methylacrylonitrile (MeAN) (Fig. 1b). As it will become clear in the remainder of this article, if the cavity mode is polarized along the forming carbon-carbon bond (corresponding to the molecular $x$-axis in Fig. 2), the cavity significantly inhibits the reaction rate, whereas the polarization of the cavity modes in the other two molecular directions catalyzes and steers the reactions to either give a major *endo* or *exo* product (Fig. 1c). The observed changes in the reactivity are due to the formation of a new light matter hybrid ground state via the quantum vacuum fluctuations. We expect our findings will further push the field of experimental and theoretical polaritonic chemistry.

## Results

### Reaction between cyclopentadiene and acrylonitrile

As first example, in Fig. 2 we show the reaction energy diagram for the Diels-Alder cycloaddition reaction between cyclopentadiene (CPD) and acrylonitrile (AN), where the left and right panels correspond to the *endo* and *exo* paths, respectively. The transition state energies (energy difference between the transition state and the reaction

**Table 1 | Reaction energy barrier (TS)[a] and reaction energy (ΔE)[b] (in kcal mol⁻¹) for Diels-Alder cycloaddition reactions of cyclopentadiene (CPD) with acrylonitrile (AN) and methylacrylonitrile (MeAN) for *endo* and *exo* paths calculated with the QED-CCSD/cc-pVDZ method for one cavity mode with frequency of 1.5 eV and light-matter coupling of 0.1 a.u**

| | | TS | *endo*:*exo*[c] | ΔE | *endo*:*exo*[d] |
|---|---|---|---|---|---|
| **CPD+AN** | | | | | |
| No cavity[e] | *endo* | 20.8 | 54:46 | −29.6 | 50:50 |
| | *exo* | 20.9 | | −29.6 | |
| *x* direction | *endo* | 26.3 | 58:42 | −26.3 | 42:58 |
| | *exo* | 26.5 | | −26.5 | |
| *y* direction | *endo* | 17.3 | 99:1 | −36.7 | 99:1 |
| | *exo* | 20.4 | | −33.6 | |
| *z* direction | *endo* | 23.1 | 1:99 | −30.4 | 0:100 |
| | *exo* | 20.3 | | −33.9 | |
| **CPD+MeAN** | | | | | |
| | | TS | *endo*:*exo*[c] | ΔE | *endo*:*exo*[d] |
| No cavity[e] | *endo* | 23.6 | 7:93 | −27.1 | 30:70 |
| | *exo* | 22.1 | | −27.6 | |
| *x* direction | *endo* | 29.0 | 6:94 | −24.3 | 30:70 |
| | *exo* | 27.4 | | −24.8 | |
| *y* direction | *endo* | 21.5 | 46:54[f] | −32.7 | 88:12 |
| | *exo* | 21.4 | | −31.5 | |
| *z* direction | *endo* | 24.7 | 1:99 | −29.5 | 1:99 |
| | *exo* | 21.6 | | −32.1 | |

[a]Calculated as the energy difference between the transition state and the reaction complex.
[b]Calculated as the energy difference between the product and the reaction complex.
[c]*endo*/*exo* ratio for kinetically controlled reaction.
[d]*endo*/*exo* ratio for thermodynamically controlled reaction.
[e]No cavity corresponds to the conventional electronic structure CCSD result.
[f]For $\lambda_y$ = 0.15 a.u. the computed *endo*/*exo* ratio is 88:12.

complex) and reaction energies (energy difference between the product and the reaction complex) along the reaction path are calculated with the coupled cluster singles and doubles (CCSD) method (also referred as 'no cavity') and the quantum electrodynamics coupled cluster singles and doubles (QED-CCSD) method with the light mode polarized in the molecular *x* (magenta), *y* (blue), and *z* (red) directions (the corresponding molecular coordinate system is shown in Fig. 2). For more details see the Methods sections. In the following, we discuss the *endo*:*exo* product ratio of the Diels-Alder cycloaddition reaction under two conditions; kinetic and thermodynamic control of the reaction. In the case of kinetically controlled reactions, the *endo*:*exo* product ratio is determined from the reaction energy barriers, whereas for thermodynamically controlled reactions the *endo*:*exo* product ratio is determined from the reaction energies. The product ratios are calculated by employing the Arrhenius equation[39] at room temperature ($T$ = 298.15 K) and are provided in Table 1. The validity of the Arrhenius equation under the strong-light matter coupling regime was experimentally confirmed in the context of a closely related electrocyclic reaction[40].

As shown in Fig. 2 and Table 1, the CCSD gas phase results show that the transition state energy is slightly lower along the *endo* path. Moreover, the computed *endo*:*exo* ratio of 54:46 for the reaction under kinetic controll is in good agreement with the experimentally determined ratio of 58:42[41], done in a liquid phase and without solvent at 298.15 K[41]. Experiments also indicate that a nonpolar solvent as well temperature from 298.15 to 373.15 K, what justify the frozen nuclei study we do in here, do not significantly affect that ratio[41]. If we perform a similar study, but where the reactants are inside an optical cavity with a fixed light matter coupling of 0.1 a.u., we found that the

barrier increases by ~5.5 kcal mol⁻¹ for both reaction paths if the cavity mode is polarized along the molecular *x* direction. Such increase in the reaction barrier corresponds to decrease in reaction rate by ~10,000 times at room temperature. Because, both paths experience nearly equal increase in the reaction barrier, the calculated *endo*:*exo* ratio changes slightly (Table 1). However, when the cavity mode is polarized in the molecular *y* direction, the reaction energy barrier along the *endo* path decreases by 3.5 kcal mol⁻¹, whereas the *exo* path decreases by only 0.5 kcal mol⁻¹. Therefore, the energy barrier for the *endo* path is lower by 3 kcal mol⁻¹ relative to the *exo* path, resulting in 99:1 *endo*:*exo* product ratio for kinetically controlled reactions. In addition, such a decrease in barrier increases the reaction rate along the *endo* path by two orders of magnitude (370 times). Lastly, for the cavity mode polarized in the molecular *z* direction, the reaction barrier along the *endo* path increases by 2.3 kcal mol⁻¹, whereas along the *exo* path decreases by 0.6 kcal mol⁻¹. This corresponds to a change in *endo*:*exo* ratio of 1:99 as well as in increase of the reaction rate for the preferred reaction path by ~3 times. We note that the transition state geometry does not change significantly under the strong light-matter interaction as discussed in Supplementary Discussion 1.

The reason for such selectivity is that the different orientations of the molecular dipole moment (indicated by orange arrow in Fig. 2) of the stationary structures (i.e., reaction complex, transition state, product) along the *endo* and *exo* paths dictate the direction in which coupling between light and a molecule is strongest, which ultimately leads to changes in reaction energy barriers and reaction energies. As shown in Fig. 2 (orange arrow) and in Supplementary Table 1, the *x* component of the molecular dipole moments between the corresponding stationary structures along the *endo* and *exo* path are of roughly the same magnitude, therefore almost no discrimination is observed for either of the path. In contrast, the *y* component of the dipole moment for the reaction complex is ~50% larger along the *endo* than along the *exo* path, resulting in the preference of the *endo* path. Similarly, the *z* component of the dipole moment for the reaction complex is greater along the *exo* path than along the *endo* path, leading to a preferred *exo* selectivity. To gain further understanding of the underlying mechanism, we have computed the individual contributions of the electronic, dipole self-energy, and dipolar coupling terms to the reaction barrier (see Supplementary Discussion 4). When the light is polarized along the *x* molecular direction, both the electronic and dipole self-energy terms contribute significantly to the change in the reaction barrier, whereas in the case when the light is polarized in the *y* and *z* molecular directions, the change in the reaction barrier is mainly due to the dipole self-energy term.

Next, we discuss the product composition under thermodynamic control (reaction energy) for the Diels-Alder cycloaddition reaction between CPD and AN. The CCSD method predicts the same reaction energy along both *endo* and *exo* paths. For thermodynamically controlled reactions, this results in equal amount of *endo* and *exo* products. For a cavity with the mode polarized in the molecular *x* direction, both reactions become less exothermic by ~3.2 kcal mol⁻¹ with a slight change of preference in favor of the *exo* product. The greatest effect of the cavity on the reaction energies is observed for the *endo* path when the cavity mode is polarized along the molecular *y* direction, where the reaction becomes more exothermic by 7.1 kcal mol⁻¹. At the same time, the reaction along the *exo* path is lowered by 4.0 kcal mol⁻¹, which indicates that for thermodynamically controlled reactions, the *endo*:*exo* product ratio is 99:1. Finally, for a cavity with light polarized in the molecular *z* direction, the reaction energy along the *endo* path is stabilized by 0.8 kcal mol⁻¹ relative to no cavity case, whereas for the *exo* path the reaction energy is lower by 4.3 kcal mol⁻¹, resulting in 0:100 *endo*:*exo* product ratio.

The left panel of Fig. 3 shows the change in content of *endo* product for a kinetically controlled Diels-Alder cycloaddition reaction between CPD and AN as the magnitude of the cavity light-matter

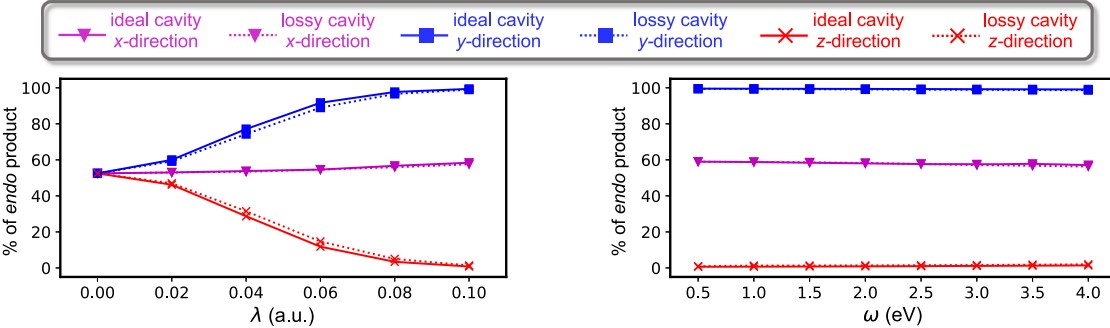

**Fig. 3 | Percentage of *endo* product for a Diels-Alder cycloaddition as a function of cavity parameters.** Percentage of *endo* product for kinetically controlled Diels-Alder cycloaddition reaction between cyclopentadiene and acrylonitrile as a function of cavity coupling strength, $\lambda$, (left panel) and cavity frequency, $\omega$, (right panel) calculated with the QED-CCSD method. The solid lines corresponds to an ideal (lossless) cavity with one cavity mode polarized along the *x* (magenta), *y* (blue), and *z* (red) molecular directions (Fig. 2). The dotted lines corresponds to a lossy (dissipative) cavity with 6000 modes and dissipation constant 1 eV. The left panel is calculated with the cavity frequency 1.5 eV, whereas the right panel is calculated with coupling strength magnitude 0.1 a.u. Source data are provided as a Source Data file.

coupling strength is increased from 0 a.u. to 0.1 a.u., while keeping the cavity frequency constant at 1.5 eV. In the case of ideal cavity with one cavity mode (solid lines) polarized along the molecular *x* direction, the content of *endo* increases by a small amount. When the cavity mode is polarized along the other two molecular directions, the content of the *endo* product changes rapidly, such that 90% of *endo* product (molecular *y* direction) or 90% of *exo* (molecular *z* direction) product is obtained for a light-matter coupling of $|\lambda| = 0.06$ a.u. The right panel of Fig. 3 shows the change in content of *endo* product for the same Diels-Alder cycloaddition reaction as the cavity frequency is increased from 0.5 to 4.0 eV, while keeping the cavity light-matter coupling strength constant at 0.1 a.u. Note that selected values of the cavity frequency are well above any molecular vibrational levels and below electronic excitations[28,29] and the selected range was motivated by the experimentally realizable values. As shown in Fig. 3, the content of the final product shows very little dependence on the cavity frequency and no resonance effect is observed, although the effect of the cavity is the greatest at low values and slowly decreases as the cavity frequency increases. The content of *endo* product for the thermodynamically controlled reaction shows qualitatively similar behavior relative to the kinetically controlled reaction as it is shown in Supplementary Fig. 2.

All results presented so far assumed an ideal (lossless) cavity. In order to approximately account for the real cavity losses, we translate the openness of the cavity into a broadening of the cavity mode in which the photon mode is broaden by explicitly including multiple discrete photon modes close in energy to the fundamental frequency of the cavity mode[42]. In this approximated treatment of a lossy cavity, the mode's width is controlled by the dissipation constant of the mirrors (a zero value of the dissipation constant corresponds to the previous lossless cavity results). This approach has been used in different contexts in quantum optics and recently within the realm of molecules coupled to a cavity in ref. 42. As shown in Fig. 3 (dotted lines), the cavity losses with a dissipation constant of 1 eV show only minor impact on the content of the *endo* product as the cavity coupling strength (left panel) and cavity frequency (right panel) increases. We found in our calculations (see Supplementary Discussion 5) that cavity losses have only minor impact on the reaction diagram for the investigated reaction.

### Reaction between cyclopentadiene and methylacrylonitrile
As second example, we investigate the Diels-Alder cycloaddition reaction between cyclopentadiene (CPD) and methylacrylonitrile (MeAN) in a cavity. Figure 4 shows the reaction energy diagram for the *endo* (left panel) and *exo* (right panel) paths. The CCSD results (no cavity) are given in black, whereas the QED-CCSD results with the

cavity mode polarized along the molecular *x*, *y*, and *z* directions are given in magenta, blue, and red, respectively.

As shown in Fig. 4 and Table 1, the reaction energy barrier calculated with the CCSD method for the *endo* path is higher by 1.5 kcal mol$^{-1}$ relative to the *exo* path. Therefore, unlike the previous reaction where the *endo* was the major product, the calculated *endo:exo* product ratio for kinetically controlled reaction is 7:93. The calculated ratio is in qualitatively good agreement with the experimentally determined ratio of 12:88[41], done in a liquid phase and without solvent at 298.15 K[41]. The greatest effect of the cavity on the reaction barrier is observed when the cavity mode is polarized along the molecular *x* direction. In that event, the reaction barrier is increased by ~5.3 kcal mol$^{-1}$, which results in the decrease in reaction rate by roughly 8000 times at room temperature for kinetically controlled reaction. In addition, the ratio of the products changes slightly relative to the no cavity case. When the cavity mode is polarized in the molecular *y* direction the reaction barrier along the *endo* path is lower by 2.1 kcal mol$^{-1}$, whereas along the *exo* path it is lowered by only 0.7 kcal mol$^{-1}$. Under these conditions, the calculated *endo:exo* ratio is 46:54. As a result, the reaction rate along the *endo* path is increased by 35 times, whereas the reaction rate along the *exo* path is increased by only three times. We note (Table 1) that when the cavity strength reaches $\lambda_y = 0.15$ a.u., the *endo:exo* product ratio is completely inverted in favor of *endo* product. In the event when the cavity mode is polarized in the molecular *z* direction, the barrier increases/decreases along the *endo*/*exo* paths, leading to formation of the *endo:exo* products ratio of 1:99.

Next, we discuss the cavity effect on the thermodynamically controlled reaction between CPD and MeAN. The CCSD method predicts that the *exo* product is thermodynamically more stable than its *endo* counterpart where the calculated *endo:exo* ratio is 30:70. If now the cavity mode is polarized in the molecular *x* direction, the reaction along both paths becomes less exothermic while preserving the ratio between *endo* and *exo* products. In the case the cavity mode is polarized along the molecular *y* direction, the effect is more pronounced along the *endo* path than along the *exo* path. This results in a ratio (88:12) where the *endo* product is more favorable. Lastly, for the cavity with light polarized along the molecular *z* direction, the *endo:exo* reaction ratio is 1:99 for thermodynamically controlled reactions.

As was the case for our first reaction, the composition of the *endo* product for kinetically and thermodynamically controlled reaction changes slowly as the cavity coupling strength increases, however this change is rapid when the cavity mode is polarized along the molecular *y* and *z* directions (for more details see Supplementary Discussion 2). The driving force responsible for this selectivity is the polarization of the cavity mode and its coupling to the molecular dipole moment (see

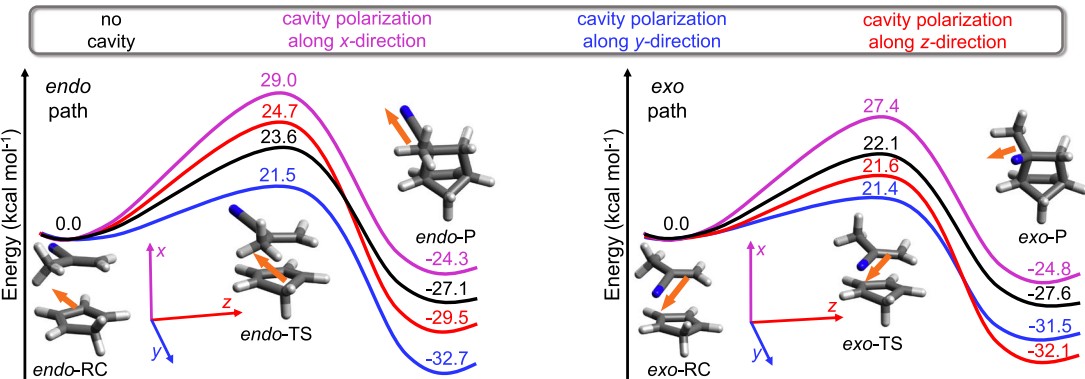

**Fig. 4 | Reaction diagram of Diels-Alder cycloaddition reaction between cyclopentadiene (CPD) and methylacrylonitrile (MeAN).** Reaction energy diagram for *endo* path (left panel) and *exo* path (right panel) calculated with the CCSD/cc-pVDZ (black) and QED-CCSD/cc-pVDZ methods. The quantum electrodynamics (QED) calculations employ the cavity frequency $\omega_{cav}$ = 1.5 eV and light-matter coupling strength $|\lambda|$ = 0.1 a.u. with one photon mode polarized in the $x$ (magenta),

$y$ (blue), and $z$ (red) molecular directions (marked in the inset of the figure). Both panels contain the images of reaction complex (RC), transition state (TS), and product (P) structures, their total dipole moment directions (orange arrow), and the molecular coordinate frame, where $x$ direction is along the forming carbon-carbon bond, and $y$ and $z$ directions lay in plane of the CPD ring. Source data are provided as a Source Data file.

the dipole moment direction in Fig. 4 and Supplementary Discussion 3). We also find that the content of the *endo* product for this reaction under kinetic and thermodynamic control does not reasonably depend on the cavity frequency (see Supplementary Discussion 2). Lastly, the calculations indicate that the cavity losses show only minor impact on the reaction energy diagram for the investigated reaction (see Supplementary Table 3).

## Discussion

In this work, we have demonstrated that an optical cavity modifies the chemical reaction of prototypical Diels-Alder cycloaddition reactions of cyclopentadiene (CPD) with acrylonitrile (AN) and methylacrylonitrile (MeAN) for both perfect and lossy cavities. Our theoretical study suggests that all the effects of the cavity observed in here are due to cavity quantum fluctuations (we are not driving the cavity by adding photons in the cavity mode). The selected Diels-Alder reactions provide an ideal platform in which one either enhances, inhibits, or controls the stereoselectivity of the products in a controlled manner for the very same reaction. We show that a cavity with the mode polarized along the forming carbon–carbon bonds increases the reaction barrier by more than 5 kcal mol$^{-1}$ for both investigated reactions without significantly impacting the *endo/exo* product ratio. Such increase of the reaction barrier corresponds to decrease in reaction rate roughly by four orders of magnitude at room temperature. A cavity with modes polarized along the molecular $y$ and $z$ directions (in the plane of the CPD ring) catalyzes both reactions along the *endo* and *exo* paths, respectively. For kinetically controlled reactions and in the case of the first reaction this corresponds to a selective production of *endo* or *exo* as a major product. In the case of the second reaction, a cavity with the mode polarized along the molecular $y$-direction gives nearly equal composition of the *endo* and *exo* products, whereas the $z$-direction oriented cavity mode gives major *exo* product. In addition to the reaction energy barrier, we show that the cavity also has a significant impact on the reaction energy. Therefore for the thermodynamically controlled reactions, major *endo* and major *exo* product is obtained with $y$ or $z$ oriented fields, respectively, in both reactions. We also note that inclusion of triple electronic excitations as well as its interactions to the photonic degrees of freedom are also possible, and it would improve the overall quantitative accuracy of the calculated reaction barriers and reaction energies as well as the effect of the cavity, but at a significantly higher cost. However, we believe that that our findings are robust with respect to those improvements in the description of electron correlation effects in the molecule and electron-photon interactions beyond the single cavity mode used here, and would not

change the main trends discussed in the present work. Finally, we show that for the investigated reactions the cavity losses do not impact the reaction dynamics in appreciable manner.

This work highlights the possibility to experimentally realize the control of the *endo/exo* stereoselectivity of Diels-Alder cycloaddition reactions. In order to achieve this experimentally, the reacting molecules must be oriented relative to the light polarization direction. Although this is not trivial to achieve, a similar control was already experimentally achieved through bottom-up nanoassembly[43] and in the context of electrostatic catalysis of a Diels-Alder reaction[44]. Moreover, the external static fields can also be used to align the molecules inside the cavity[45]. As experimentally confirmed, application of external static fields would increase the reaction rate only if the electric field is aligned with the reaction axis, otherwise leaving the reaction rate unchanged[46]. Because of that, observed *endo/exo* selectivity due to strong light-matter would not be affected by the presence of the external electric fields. Although the present study have focused on a single-molecule reaction inside a cavity, it represents the necessary starting point for exploiting other effects such as collectivity[25,38], because the observed changes in chemical reactivity and selectivity can only be enhanced in the large collective limit if the cavity is able to modify the energy landscape of a single-molecule reaction. Another importance of our work is that it represents the first computational study of cavity enhancement in click chemistry reactions. Because the click chemistry reactions are one of the most powerful tools in drug discovery, chemical biology, and proteomic applications[2], we believe that this work will further stimulate theoretical investigations and experiments of other click chemical reactions under the strong light-matter regime.

## Methods
### Theoretical background

The interaction between quantized light and a single molecule or collection of molecules confined to a cavity with $N$ cavity modes can be formally described with the Pauli-Fierz Hamiltonian[21]. Under the dipole approximation, in the length gauge[47], and in the coherent state basis[20,21,30], this light-matter interacting Hamiltonian (in atomic units) takes the following form:

$$\hat{H} = \hat{H}^e + \sum_{\alpha}^{N} \left( \omega_\alpha b_\alpha^\dagger b_\alpha - \sqrt{\frac{\omega_\alpha}{2}}(\boldsymbol{\lambda}_\alpha \cdot \Delta \boldsymbol{d})(b_\alpha^\dagger + b_\alpha) + \frac{1}{2}(\boldsymbol{\lambda}_\alpha \cdot \Delta \boldsymbol{d})^2 \right).$$

$$(1)$$

In this equation, $\hat{H}^e$ corresponds to the electronic Hamiltonian within the Born-Oppenheimer approximation (non-Born-Oppenheimer effects can also be incorporated[48–50]), which is defined as:

$$\hat{H}^e = h_q^p a_p^q + \frac{1}{2} g_{rs}^{pq} a_{pq}^{rs}. \tag{2}$$

The electronic Hamiltonian is expressed in terms of the second-quantized electronic excitation operators $a_{p_1 p_2 \ldots p_n}^{q_1 q_2 \ldots q_n} = a_{q_1}^\dagger a_{q_2}^\dagger \ldots a_{q_n}^\dagger a_{p_n} \ldots a_{p_2} a_{p_1}$, where $a_e^\dagger / a$ are fermionic creation/annihilation operators. Moreover, $h_q^p = \langle q | \hat{h} | p \rangle$ and $g_{rs}^{pq} = \langle rs | pq \rangle$ are matrix elements of the core electronic Hamiltonian and the two-electron repulsion operator, respectively. Throughout this work, the $i, j, k, l, \ldots$ indices denote occupied electronic spin orbitals, $a, b, c, d, \ldots$ denote unoccupied electronic spin orbitals, and $p, q, r, s, \ldots$ indices denote general electronic spin orbitals. We adopt Einstein's notation where the summation over the repeated indices is assumed. The second term in Eq. (1) denotes the photonic Hamiltonian for $N$ cavity modes of frequency $\omega_\alpha$. The operators $b_\alpha^\dagger / b_\alpha$ are bosonic creation/annihilation operators that create/destroy a photon in cavity mode $\alpha$. The third term describes the coupling between the electronic and photonic degrees of freedom in the dipole approximation. Within this term, the $\Delta d = d - \langle d \rangle$ is the dipole fluctuation operator that describes the change of the molecular dipole moment operator with respect to its expectation value, and $\lambda_\alpha$ is the light-matter coupling strength of the $\alpha$-th cavity mode. The last term represents the dipole self energy that ensures the origin invariance of the Hamiltonian and Hamiltonian's boundness from below[47,51].

To describe the interaction of quantized light with a molecule, we need to solve the time-independent Schrödinger equation

$$\hat{H} | \Psi_{\text{QED-CC}} \rangle = E_{\text{QED-CC}} | \Psi_{\text{QED-CC}} \rangle, \tag{3}$$

where $E_{\text{QED-CC}}$ is the quantum electrodynamics coupled cluster (QED-CC) energy of a system confined to a cavity, and $| \Psi_{\text{QED-CC}} \rangle$ is the QED-CC ground state wave function ansatz[30–33] expressed as

$$| \Psi_{\text{QED-CC}} \rangle = e^{\hat{T}} | 0^e 0^{\text{ph}} \rangle. \tag{4}$$

In this equation, $\hat{T}$ is the cluster operator[52] that accounts for important correlations between quantum particles (i.e., electrons and photons) by generating excited configurations from the reference configuration $| 0^e 0^{\text{ph}} \rangle = | 0^e \rangle \otimes | 0^{\text{ph}} \rangle$, where $| 0^e \rangle$ is an electronic Slater determinant and $| 0^{\text{ph}} \rangle$ is a photon-number state. In this work, we use the QED-CC method in which the cluster operator is truncated to include up to single and double electronic excitations and single photon creation, along with interactions between single electron with single photon as

$$\hat{T} = t_a^i a_i^a + \frac{1}{4} t_{ab}^{ij} a_{ij}^{ab} + \sum_\alpha t_\alpha b_\alpha^\dagger + \sum_\alpha t_{a,\alpha}^i a_i^a b_\alpha^\dagger. \tag{5}$$

Here $a_i^a$ and $a_{ij}^{ab}$ are single and double electronic excitation operators, respectively. The $\{t_a^i, t_{ab}^{ij}, t_\alpha, t_{a,\alpha}^i\}$ are set of unknown wave function parameters that are determined using the projective technique[30]. We will refer to this method as QED-CCSD.

Due to its simplicity, the QED-CCSD method has a similar computational cost as the conventional electronic structure CCSD method. Because it is an approximation to other QED-CC methods investigated previously[30–34,53], this method has also been benchmarked against

more accurate method in which the cluster operator $\hat{T}$ is defined as

$$\begin{aligned} \hat{T} = {} & t_a^i a_i^a + \frac{1}{4} t_{ab}^{ij} a_{ij}^{ab} + \sum_\alpha t_\alpha b_\alpha^\dagger + \sum_\alpha t_{a,\alpha}^i a_i^a b_\alpha^\dagger \\ & + \frac{1}{4} \sum_\alpha t_{ab,\alpha}^{ij} a_{ij}^{ab} b_\alpha^\dagger + \frac{1}{2} \sum_{\alpha\beta} t_{\alpha\beta} b_\alpha^\dagger b_\beta^\dagger \\ & + \frac{1}{2} \sum_{\alpha\beta} t_{a,\alpha\beta}^i a_i^a b_\alpha^\dagger b_\beta^\dagger + \frac{1}{8} \sum_{\alpha\beta} t_{ab,\alpha\beta}^{ij} a_{ij}^{ab} b_\alpha^\dagger b_\beta^\dagger. \end{aligned} \tag{6}$$

In this method, up to two electrons are interacting with up to two photons, therefore this method is denoted as QED-CCSD-22[33]. Programmable expressions and the working code for the QED-CCSD-22 method with single photon mode is provided in ref. 33. Note that the QED-CCSD method is simply obtained by omitting relevant terms from the QED-CCSD-22 method. The performance of the QED-CCSD method against a more accurate but computationally costly QED-CCSD-22 method is provided in the Supplementary Discussion 6. The level of accuracy for the QED-CCSD method is adequate to support the conclusions about catalytic control of the Diels-Alder reaction discussed in this work.

## Specifics of the cavity
Throughout this work, we work in the strong light-matter coupling regime by considering a $|\lambda| = 0.1$ a.u.[27,33,54–56]. Recent advances in the cavity design are able to realize the strong light-matter coupling to within this value such as achieved in case of the Landau polaritons[29], distributed Bragg reflector cavities[28], and in plasmonic nanocavities[43]. Boosting the light-matter coupling can be achieved by exploiting the collectivity effect[17,25], by taking advantage of interactions mediated by ensemble of nearby emitters[57], by the multimode cavities[58], or by employing the picocavities[59]. We note that for description of the strong light-matter effects in the nanocavities and picocavities, one must go beyond the dipole approximation. The contributions beyond the dipole approximation may be relevant for obtaining the quantitatively accurate results, however the qualitative results that we show in here will not be modified. In addition, for other cavity setups the dipole term would be the dominant one even in quantitative terms.

## Computational details
The QED-CC methods for lossless (ideal) and lossy (dissipative) cavities have been implemented in an in-house developmental version of the Psi4NumPy quantum chemistry software[60]. The source code for the QED-CC methods is available in ref. 33. The QED-CC methods were used to calculate the reaction energy diagrams for two different Diels-Alder cycloaddition reactions. All calculations were performed on the geometries optimized at the conventional electronic MP2/cc-pVDZ[61] level of theory using the Q-Chem quantum chemistry software[62]. The nature of optimized geometries (i.e., reaction complexes, transition states, and products) were confirmed by performing the harmonic frequency analysis, where transition state structures have one imaginary vibrational frequency, and reaction complex and product structures have zero imaginary vibrational frequencies. The optimized structures are provided within the Supplementary Data file. All of the CCSD and QED-CC calculations were performed with the cc-pVDZ basis set[61]. In our studies, the reaction energy barriers and reaction barriers are calculated with cavity parameters $|\lambda| = 0.1$ a.u. and with one photon mode of frequency $\omega = 1.5$ eV. Along the reaction path, orientation of the structures was kept constant such that the forming carbon-carbon bond is oriented along $x$-direction for all structures, whereas the front three carbon atoms (the carbon in the middle is sp$^3$ hybridized) form the $yz$ plane.

**Reporting summary**

Further information on research design is available in the Nature Portfolio Reporting Summary linked to this article.

## Data availability

The authors declare that all data that support the findings of this study are available within the Supplementary Information file, on the public repository https://github.com/fabijan5/qed-diels-alder[63], or from the authors upon request. Source data are provided with this paper.

## Code availability

The code used for obtaining the numerical results is available from the corresponding authors or is available free of charge from the Supplementary Material of ref. 33.

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

## Acknowledgements

A.R. acknowledge financial support from the Cluster of Excellence 'CUI: Advanced Imaging of Matter'- EXC 2056 - project ID 390715994 and SFB-925 "Light induced dynamics and control of correlated quantum systems" - project 170620586 of the Deutsche Forschungsgemeinschaft (DFG) and Grupos Consolidados (IT1453-22). A.R. also acknowledge support from the Max Planck-New York Center for Non-Equilibrium Quantum Phenomena. F.P., R.L.S., and A.R. acknowledge that The Flatiron Institute is a division of the Simons Foundation.

## Author contributions

F.P. and A.R. conceived the project. F.P. proposed the investigated applications, oversaw the project, wrote the code, prepared the figures, and wrote the first draft. R.S. and F.P. obtained and evaluated the data. All authors participated in analyzing and discussing the results, and editing the manuscript.

## Competing interests

The authors declare no competing interests.
