## [Peer Review File · Nature Communications]

Computational study on the catalytic control of endo/exo Diels-Alder reactions by cavity quantum vacuum fluctuationsREVIEWER COMMENTS

Reviewer #1 (Remarks to the Author):

The authors explored the possibility to employ an optical cavity to control the selectivity of a Diels-Alder reaction. They showed that the selectivity of endo and exo-products of a reaction can be controlled by tuning the polarization of the cavity mode. Recently chemistry in cavities is emerging as a new platform to manipulate chemical reactions, both ground state and excited states. This is another interesting illustration of controlling a ground-state chemical reaction by optical cavities.

Nevertheless, the conclusions of the reaction rates are based solely on the change of energy profile induced by the cavity under ideal conditions. This raises some concerns.

If they address these issues the paper will make a nice contribution to NCOMMS

1. First, this is not directly a polariton effect. The electronic polaritonic states are reflected in the excited-state PES, but the reported reaction is a ground state reaction. This should be clarified. Also, what is the main contribution to the energy shift of the ground state PES? An analysis would be very useful.

2. The conditions to observe the proposed cavity effects are very difficult to achieve experimentally. Does it need perfectly aligned molecule?

3. Is the Arrhenius equation still valid for polaritonic systems as the cavity should decay much faster than the reaction?

4. How is the transition state in the cavity determined? It is interesting to see how the transition state geometry depends on the cavity parameters.

Reviewer #2 (Remarks to the Author):

The paper "Catalysis controlled by cavity quantum vacuum fluctuations: the case of endo/exo Diels-alder reaction" is a theoretical study based on a recently developed method QED-CC that extends the coupled-clusters description of many-body electronic states to include a photonic part. The authors then apply the theory to describe a cycloaddition reaction whose outcome depends on the orientation of an electric field of an optical mode of a cavity.

The introduction of the paper states that "in order to understand and explain those [electronic polariton effects] experiments, ab initio methods in which electrons, nuclei, and photons are treated on equal footing are required". This is a strong statement that may be generally hard to prove (there may be theoretical models implementing light-matter interactions etc. in a different way, yet be adequate to describe the cycloaddition reaction studied here). I would therefore suggest down-scaling such strong statements unless a proof of the opposite is given.

I find one point of the paper particularly concerning. It is the mode volume of the cavity considered ($V \ll 1 \text{ nm}^3$). For such a small mode volume (theoretically achievable maybe just in special plasmonic cavities), the dipolar approximation assumed by the author is bound to fail, and the validity of the theory is therefore questionable. Moreover, in plasmonic cavities the molecule is going to be very close to metallic surfaces that will significantly influence the molecular properties by e.g. image-charge effects or direct orbital overlap between the molecule and the metal. In such a case, the theory applied by the authors would likely be

inadequate. This concern should be carefully addressed by the authors.

Similarly, the claim of the conclusions that the authors "demonstrated" that the chemical reaction is modified by quantum fluctuations of EM field is a bit overstated due to the purely theoretical character of the work. At best, the work can suggest such an effect, especially then if so unrealistic (or borderline unrealistic) conditions are considered.

Despite this, I find the study an interesting contribution to the treatment of light-matter coupling and its potential effects on molecular reactivity interesting and publishable elsewhere (not in Nat. Commun.).

I would also like to see how the effects of EM-field fluctuations would influence the reaction in a realistic scenario containing large numbers of randomly oriented molecules. Would it still be possible to steer the reaction? I find the single-molecular case with a fixed geometrical orientation somewhat artificial.

I also have a question related to the effect of the dipole-moment orientation of the molecule with respect to the cavity field. How does this effect exactly arise from the light-matter coupling Hamiltonian (eq.1) of the paper? What is the exact mechanism at play?

Reviewer #3 (Remarks to the Author):

In this theoretical study, it is shown that the reaction products of a couple of Diels-Alder reactions under electronic strong coupling depends on the orientation of the molecules relative to the cavity field. This is yet another demonstration of the potential of polaritonic chemistry and as such it will be of broad interest. The only troublesome points are the following: the proposed reactant molecules have electronic transitions only in the deep UV (ca. 5 eV for cyclopentadiene and even higher energies for the nitriles) not at 1.5 eV which seems to be the basis for the calculations (Fig. 3 left panel and discussion on pages 4 and 5). The authors should say more clearly what they are coupling and demonstrate that they are in the strong coupling regime. Secondly, the tiny quantization volumes are not realistic for any practical condition. This should be discussed.

Response to Reviewers: NCOMMS-22-37279-T

The reviewer comments are shown in *italic* and our reply is in normal font.

Reviewer: 1

Comments:

The authors explored the possibility to employ an optical cavity to control the selectivity of a Diels-Alder reaction. They showed that the selectivity of endo and exo-products of a reaction can be controlled by tuning the polarization of the cavity mode. Recently chemistry in cavities is emerging as a new platform to manipulate chemical reactions, both ground state and excited states. This is another interesting illustration of controlling a ground-state chemical reaction by optical cavities.

Nevertheless, the conclusions of the reaction rates are based solely on the change of energy profile induced by the cavity under ideal conditions. This raises some concerns.

If they address these issues the paper will make a nice contribution to NCOMMS.

We thank the reviewer for the thoughtful comments and we have modified the article to clarify the main issues raised by the reviewer. Please see our comments and corresponding modifications in the manuscript below.

Comment 1:

First, this is not directly a polariton effect. The electronic polaritonic states are reflected in the excited-state PES, but the reported reaction is a ground state reaction. This should be clarified. Also, what is the main contribution to the energy shift of the ground state PES? An analysis would be very useful.

We thank the reviewer for this comment. The observed changes of the ground state PES for the studied chemical process are due to the quantum vacuum fluctuations of the cavity. This fact is indicated in the title as "Catalysis in Click Chemistry Reactions Controlled by Cavity Quantum Vacuum Fluctuations" where the driving forces are quantum vacuum fluctuations and not the excited state electronic polaritonic states. To additionally clarify that fact and to make it more clear, we have added one sentence to the Introduction on Page 2 of the manuscript:

"The observed changes in the reactivity are due to the formation of a new light matter hybrid ground state via the quantum vacuum fluctuations."

Moreover, to address the reviewer's comment regarding the analysis, we have calculated the individual energy contributions (electronic, dipole self-energy, and dipolar coupling) for the QED-CCSD method and we have added the following statement on Page 4 of the manuscript:

"To gain further understanding of the underlying mechanism, we have computed the individual contributions of the electronic, dipole self-energy, and dipolar coupling terms to the reaction barrier (see Section 6 of the Supplemental Information). When the light is polarized along the x molecular direction, both the electronic and dipole self-energy terms contribute significantly to the change in the reaction barrier, whereas in the case when the light is polarized in the y and z molecular directions, the change in the reaction barrier is mainly due to the dipole self-energy term."

as well as an additional section "Energy Contributions Breakdown" in the Supplemental Information with accompanying Supplemental Table III as follows:

"Supplemental Table III shows the energy contributions breakdown for the two studied DA cycloaddition reactions. The energy contributions for the QED-CCSD method with one cavity mode are: electronic $\left(\langle 0^e 0^{\text{ph}} | \hat{H}^e | 0^e 0^{\text{ph}} \rangle + \bar{g}_{ij}^{ab} \times (0.25 \cdot t_{ab}^{ij} + 0.5 \cdot t_a^i t_b^j)\right)$, dipole self-energy $\left(\langle 0^e 0^{\text{ph}} | \frac{1}{2} (\lambda \cdot \Delta \mathbf{d})^2 | 0^e 0^{\text{ph}} \rangle + ((\lambda \cdot \mathbf{d})_{ij}^{ab})^2 \times (0.25 \cdot t_{ab}^{ij} + 0.5 \cdot t_a^i t_b^j)\right)$, and dipolar coupling $\left((\lambda \cdot \mathbf{d})_i^a \times (t_{a,1}^i + t_a^i t_1)\right)$. The $\langle 0^e 0^{\text{ph}} | \hat{H}^e | 0^e 0^{\text{ph}} \rangle$ and $\langle 0^e 0^{\text{ph}} | \frac{1}{2} (\lambda \cdot \Delta \mathbf{d})^2 | 0^e 0^{\text{ph}} \rangle$ corresponds to electronic and dipole self-energy contributions, respectively, calculated with the QED-Hartree-Fock method. The calculations employ cavity parameters $\lambda = 0.1$ a.u. and $\omega = 1.5$ eV. The Supplemental Table III indicate that the electronic energy contributes the most to the energy, whereas the changes in energy due to the cavity are mainly due to the dipole self-energy contributions. These changes in the dipole self-energy are due to the quantum vacuum fluctuations that are the main driving forces for the reactions inside a cavity."

Comment 2:

The conditions to observe the proposed cavity effects are very difficult to achieve experimentally. Does it need perfectly aligned molecule?

We thank the reviewer for raising this important question concerning the experimental realisation of the present findings. We want to note that a perfect alignment is not needed for the effect described here to be active, but a relative good alignment needs to be present. In a random molecular sample the effect discussed in this work would not be statistically relevant. Therefore, in order to observe the effects reported in this work, it is necessary that the molecules are partially aligned during the reaction. We agree that this is a nontrivial experimental task, but it is however a possible task. This has already been achieved by Baumberg and his co-workers in which their experimental techniques are capable of controlling the relative orientation of a molecular system to the cavity field polarization through bottom-up nanoassembly [1]. Moreover, another technique was also developed for controlling the orientation of the molecule in order to study the electrostatic catalysis of a Diels–Alder reaction [2]. Additionally, the molecular alignment can also be achieved by employing the static electric fields [3]. To address more specifically this point, we have included three sentences in the Summary and Conclusion section on Page 7 of the manuscript:

"In order to achieve this experimentally, the reacting molecules must be oriented relative to the light polarization direction. Although this is not trivial to achieve, a similar control was already

experimentally achieved through bottom-up nanoassembly [1] and in the context of electrostatic catalysis of a Diels–Alder reaction [2]. Moreover, the external static fields can also be used to align the molecules inside the cavity [3]."

Comment 3:

Is the Arrhenius equation still valid for polaritonic systems as the cavity should decay much faster than the reaction?

We thank the reviewer for asking this important question. In a recent work, Ebbesen and co-workers have explored the effect of cavity in an electrocyclization reaction for which is shown that the Eyring/Arrhenius equation is valid. [4] Because the Diels-Alder reaction is a rather similar reaction to the electrocyclic reaction due to the fact that both are concerted pericyclic reactions, it is expected that that the Eyring/Arrhenius equation is valid for polaritonic states, and should describe the Diels-Alder reaction inside the cavity. To address the reviewer's comment, we have added one sentence on Page 3 of the manuscript:

"The validity of the Arrhenius equation under the strong-light matter coupling regime was experimentally confirmed in the context of a closely related electrocyclic reaction [4]."

Comment 4:

How is the transition state in the cavity determined? It is interesting to see how the transition state geometry depends on the cavity parameters.

We thank the reviewer for asking this important question. As discussed in the Section 3 (Computational Details) of the Supplemental Information, all of the calculations were performed on the geometries (reaction complexes, transition states, products) optimized with the conventional electronic structure MP2/cc-pVDZ method in gas phase. In this work, we have not done the structural relaxation at the QED-CC level and we have kept the path and geometries in the gas phase to highlight what is the effect on that path. Such geometry relaxation under the light-matter coupling can be done within the QEDFT framework or at the QED-CC level but this would require further developments that are beyond the scope of this work.

To address the issue about how much the geometry of the transition state is changed due to light-matter coupling issue, we have performed the Intrinsic Reaction Coordinate (IRC) calculation at the MP2/cc-pVDZ level of theory that provided us with the path that connects reaction complex and product with transition state. This is done for both the *endo* and *exo* paths. Although this PES does not exactly corresponds to the QED-CCSD PES, we assume that it is a good approximation of it, since reaction coordinate around the transition state mainly corresponds to the change in distance between CPD and AN. Therefore, we have taken one point from each side of the transition state and performed the CCSD and QED-CCSD calculations on these geometries. The results are summarized in Fig. below, indicating that the maximum energy that is corresponding to the transition state is the same for both CCSD and QED-CCSD with all three light polarization directions. Based on that, we do not expect that the transition state will dramatically change inside the cavity for the given value

of the light-matter coupling. Additionally, because the calculations employ very large value of the light-matter coupling strength ($\lambda = 0.1$ a.u.), the changes in geometries relative to the light-matter coupling strength would be even less pronounced for the lower values of λ .

Therefore in order to accommodate the reviewer's remark, we have added the following sentence on Page 4 of the manuscript:

"We note that the transition state geometry does not change significantly under the strong light-matter interaction as discussed in Section 3 of the Supplemental Information."

Additionally, we have included the aforementioned Figure in Section 3 of the Supplemental Information as well as the accompanying discussion:

"All of the reported calculations in this work assume that the molecular geometry does not change significantly due to strong light-matter interaction. This is however an approximation and in order to check its validity, we have performed the IRC calculation along the *endo* and *exo* paths at the MP2/cc-pVDZ level of theory that provided us with the path that connects the reaction complex and the product with the transition state. Although this path may not correspond to the CCSD or QED-CCSD path, but since reaction coordinate around the transition state mainly corresponds to the change in distance between CPD and AN, this is a good assumption. Next, we have taken one point from each side of the transition state and performed the CCSD and QED-CCSD calculations on these geometries as given in Figure 1. The Figure 1 shows that the maximum energy that is corresponding to the transition state is the same for both CCSD and QED-CCSD with all three light polarization directions. Based on that, we do not expect that the transition state will dramatically change inside the cavity for the selected value of the light-matter coupling. Additionally, because the calculations employ relatively large value of the light-matter coupling strength ($|\lambda| = 0.1$ a.u.), the changes in geometries with respect to the light-matter coupling strength will be less pronounced for the lower values of $|\lambda|$."

We thank the reviewer for the very helpful and positive feedback and we hope that the manuscript with the additional information is now suitable for publication in *Nature Communications*.

Reviewer: 2

Comments:

The paper "Catalysis controlled by cavity quantum vacuum fluctuations: the case of endo/exo Diels-alder reaction" is a theoretical study based on a recently developed method QED-CC that extends the coupled-clusters description of many-body electronic states to include a photonic part. The authors then apply the theory to describe a cycloaddition reaction whose outcome depends on the orientation of an electric field of an optical mode of a cavity.

We thank the reviewer for the thoughtful comments and we have modified the article to clarify the main issues raised by the reviewer. Please see our comments and corresponding modifications in the manuscript below.

Comment 1:

The introduction of the paper states that "in order to understand and explain those [electronic polariton effects] experiments, ab initio methods in which electrons, nuclei, and photons are treated on equal footing are required". This is a strong statement that may be generally hard to prove (there may be theoretical models implementing light-matter interactions etc. in a different way, yet be adequate to describe the cycloaddition reaction studied here). I would therefore suggest down-scaling such strong statements unless a proof of the opposite is given.

We appreciate the useful reviewer's comment and we agree that this statement may be too strong. When writing this statement, our intention was to emphasize that in general, the full quantum mechanical treatment of a system leads to the exact description of a process under study. It is well known that description of the Diels-Alder reactions in the ground state can be accomplished within the Born-Oppenheimer approximation where only electrons are treated quantum mechanically and the nuclear coordinates enter the description parametrically. Thus the calculated reaction barriers and reaction energies for different Diels-Alder reactions with various conventional quantum chemical methods (under the Born-Oppenheimer approximation) are in excellent agreement with the experimentally determined values as shown in Ref. [5]. Therefore, we expect that the Born-Oppenheimer approximation is also adequate for description of the Diels-Alder reactions under the strong light-matter regime. However, for the regimes in which the energies of electrons and photons are comparable, such as in description of the exciton polaritons considered in this work, the electrons and photons should be treated quantum mechanically on the equal footing which is achieved in the present study. To address the reviewer's comment, we have replaced the concerning sentence on Page 1 of the manuscript with the following one:

"Naturally, in order to understand and explain these experiments that operate in the exciton-polariton regime, *ab initio* methods in which electrons and photons are treated quantum mechanically on equal footing are required."

Comment 2:

I find one point of the paper particularly concerning. It is the mode volume of the cavity considered ($V \ll 1\text{nm}^3$). For such a small mode volume (theoretically achievable maybe just in special plasmonic cavities), the dipolar approximation assumed by the authors is bound to fail, and the validity of the theory is therefore questionable. Moreover, in plasmonic cavities the molecule is going to be very close to metallic surfaces that will significantly influence the molecular properties by e.g. image-charge effects or direct orbital overlap between the molecule and the metal. In such a case, the theory applied by the authors would likely be inadequate. This concern should be carefully addressed by the authors..

We thank the reviewer for raising a concern about the cavity volume. It is correct that our description is neglecting the interactions of reacting molecules with nearby metallic interfaces in the case of the nanoplasmonic cavities, and we agree that for complete description of such systems, the electronic structure of the nanoplasmonic cavity should be taken into account. However, the interaction of molecules with plasmonic degrees of freedom is not the focus of the present study but, rather the cavities in which these effects are not present. We would like to stress out that for a given volume of the cavity (much larger than the one mentioned by the referee and where the dipole approximation will be justified), the light-matter coupling used in this work corresponds to an effective "single mode" description of a multimode cavity. In the case of the methods employed herein, the light-matter strength can be greatly enhanced by multi-mode cavities via $\lambda_{\text{eff}}^2 = \sum_n \lambda_n^2$ where λ_n is the coupling strength of the n -th mode. Therefore, in the multimode cavities that were experimentally realized by Lev and co-workers [6], the light-matter coupling can be greatly enhanced by combining multiple cavity modes while keeping relatively large cavity volumes. In order to address the reviewer's concern, we have replaced a paragraph on Page 3 of the manuscript:

"Finally, we discuss in more detail the specifics of the cavity. The light-matter coupling strength in Eq. 1. is defined by $|\lambda_\alpha| = 1/\sqrt{\epsilon_0 V_\alpha^{\text{eff}}}$. Here, ϵ_0 is the permittivity of the vacuum and V_α^{eff} is the effective quantization volume [7] that depends on the nature of the cavity. In fact, recent advances in optical cavity design [8, 9, 10, 11] realize effective volumes as low as 0.15 nm^3 [12] which corresponds to strong light-matter coupling strength above 0.1 a.u and enable the possibility of controlling the molecular orientation inside the cavity [1]. Throughout this work, we work in the strong light-matter coupling regime by considering a $\lambda = 0.1$ a.u. that corresponds to $V_{\text{eff}} = 0.19 \text{ nm}^3$ [13]. This value of the coupling constant is within the experimental range [14]. Further details are provided in the Section 3 of the Supplemental Information."

with the following paragraph:

"Finally, we discuss in more detail the specifics of the cavity. The light-matter coupling strength in Eq. 1. is defined by $|\lambda_\alpha| = 1/\sqrt{\epsilon_0 V_\alpha^{\text{eff}}}$. Here, ϵ_0 is the permittivity of the vacuum and V_α^{eff} is the effective quantization volume [7] of a single mode cavity. Recent advances in the multi-mode cavity design [6, 8, 9, 10, 11] are able to realize the strong light-matter coupling strength above 0.1 a.u. and also enable the possibility of controlling the molecular orientation inside the cavity [1]. In the

multi-mode cavities [6], the light-matter coupling strength can be greatly enhanced by increasing the number of cavity modes via $\lambda_{\text{eff}}^2 = \sum_n \lambda_n^2$, where λ_n is the coupling strength of the n -th cavity mode. Throughout this work, we work in the strong light-matter coupling regime by considering a $|\lambda| = 0.1$ a.u. [15, 16, 17, 18, 19]."

Comment 3:

Similarly, the claim of the conclusions that the authors "demonstrated" that the chemical reaction is modified by quantum fluctuations of EM field is a bit overstated due to the purely theoretical character of the work. At best, the work can suggest such an effect, especially then if so unrealistic (or borderline unrealistic) conditions are considered. .

We thank the reviewer for raising this point. The reason for such claim is that the observed changes in the reactivity of the studied Diels-Alder reactions are due modification of the ground state PES by the cavity quantum fluctuations. The strength of those fluctuations depend on the light matter coupling, that is possible to reach. To address this point, we have included a few sentences on Pages 2 and 6 to the manuscript:

"The observed changes in the reactivity are due to the formation of a new light matter hybrid ground state via the quantum vacuum fluctuations."

and:

"In this work, we have demonstrated that an optical cavity modifies the chemical reaction of prototypical Diels–Alder cycloaddition reactions of cyclopentadiene (CPD) with acrylonitrile (AN) and methylacrylonitrile (MeAN) for both perfect and lossy cavities. Our theoretical study suggests that all the effects of the cavity observed in here are due to cavity quantum fluctuations (we are not driving the cavity by adding photons in the cavity mode)."

Comment 4:

I would also like to see how the effects of EM-field fluctuations would influence the reaction in a realistic scenario containing large numbers of randomly oriented molecules. Would it still be possible to steer the reaction? I find the single-molecular case with a fixed geometrical orientation somewhat artificial.

We thank the reviewer for raising this important point about interest in collectivity effect and how would this affect the observed results. We are in a complete agreement with the reviewer that investigating this effect would bring us a step closer to the realistic scenario. First of all, we would like to note that in a random molecular sample the effects discussed in this paper would not be statistically relevant. To observe the catalytic/inhibition effect studied in this paper, we need first to have the reactants aligned within a reasonable degree of alignment and then the quantum fluctuations will steer the single molecular reaction (as well as collective). However, due to the very high computational cost of the quantum electrodynamics coupled cluster (QED-CC) method

employed in this work, we are limited to only one molecule. In the case we want to study a collection of molecules, we would be forced to trade the state-of-the-art QED-CC method with an empirically motivated method or where photons are treated classically which would miss the important effects. Recent findings by Reichman and his co-workers shows that when a molecule is coupled to an optical cavity, it is essential to treat the problem quantum mechanically in order to obtain a quantitative account of alterations to reactivity [20]. In other words, introducing approximations to our QED-CC method the effect due to strong light-matter would be minute or negligible. Therefore, the aim of this work is to study the strong light-matter effect on a single molecule with the currently most accurate method available, since we believe (due to theoretical evidences [21, 22]), that the effect we are now observing in the case of single molecule could be enhanced due to collectivity only if a single molecule case shows the appropriate behaviour. To address the reviewer's question whether would it still be possible to steer the reaction if multiple molecules are involved, we have highlighted the sentence in the conclusion section on Page 7 to the manuscript:

"Although the present study have focused on a single-molecule reaction inside a cavity, it represents the necessary starting point for exploiting other effects such as collectivity [21, 22], because the observed changes in chemical reactivity and selectivity can only be enhanced in the large collective limit if the cavity is able to modify the energy landscape of a single-molecule reaction."

This claim can not be currently investigated due to the sheer cost of the QED-CC method. This investigation represent a scientifically important and interesting future research direction. Our future research direction will be aimed toward development of computationally efficient and linear scaling [23] QED-CC method suitable for molecular assemblies comprised of a few thousand of atoms.

Moreover, regarding the reviewer's comment about the fact that the single-molecular case with a fixed geometrical orientation somewhat artificial, we have added a few sentences in the conclusion to address this point. Namely, the single molecule processes with the fixed geometry has already been achieved by Baumberg and his co-workers in which their experimental techniques are capable of controlling the relative orientation of the molecular system to the cavity field polarization through bottom-up nanoassembly [1]. Moreover, another technique was also developed for controlling the orientation of the molecule in order to study the electrostatic catalysis of a Diels–Alder reaction [2]. Additionally, the molecular alignment can also be achieved by employing the static electric fields [3]. The following sentence was added on Page 7 of the manuscript:

"In order to achieve this experimentally, the reacting molecules must be oriented relative to the light polarization direction. Although this is not trivial to achieve, a similar control was already experimentally achieved through bottom-up nanoassembly [1] and in the context of electrostatic catalysis of a Diels–Alder reaction [2]. Moreover, the external static fields can also be used to align the molecules inside the cavity [3]."

Comment 5:

I also have a question related to the effect of the dipole-moment orientation of the molecule with

respect to the cavity field. How does this effect exactly arise from the light-matter coupling Hamiltonian (eq.1) of the paper? What is the exact mechanism at play? .

We thank the reviewer for asking this excellent question. Namely, the changes in the PES are mainly due to fluctuations of the dipole self-energy term (the last term in the Hamiltonian defined in Eq. 1.). And in order to address the reviewer's question, we have calculated the individual energy contributions (electronic, dipole self-energy, and dipolar coupling) for the QED-CCSD method and we have added the following statement on Page 4 of the manuscript:

"To gain further understanding of the underlying mechanism, we have computed the individual contributions of the electronic, dipole self-energy, and dipolar coupling terms to the reaction barrier (see Section 6 of the Supplemental Information). When the light is polarized along the x molecular direction, both the electronic and dipole self-energy terms contribute significantly to the change in the reaction barrier, whereas in the case when the light is polarized in the y and z molecular directions, the change in the reaction barrier is mainly due to the dipole self-energy term."

as well as an additional section "Energy Contributions Breakdown" in the Supplemental Information with accompanying Supplemental Table III as:

"Supplemental Table III shows the energy contributions breakdown for the two studied DA cycloaddition reactions. The energy contributions for the QED-CCSD method with one cavity mode are: electronic $\left(\langle 0^e 0^{ph} | \hat{H}^e | 0^e 0^{ph} \rangle + \bar{g}_{ij}^{ab} \times (0.25 \cdot t_{ab}^{ij} + 0.5 \cdot t_a^i t_b^j)\right)$, dipole self-energy $\left(\langle 0^e 0^{ph} | \frac{1}{2} (\lambda \cdot \Delta \mathbf{d})^2 | 0^e 0^{ph} \rangle + ((\lambda \cdot \mathbf{d})_{ij}^{ab}) \times (0.25 \cdot t_{ab}^{ij} + 0.5 \cdot t_a^i t_b^j)\right)$, and dipolar coupling $\left((\lambda \cdot \mathbf{d})_i^a \times (t_{a,1}^i + t_a^i t_1)\right)$. The $\langle 0^e 0^{ph} | \hat{H}^e | 0^e 0^{ph} \rangle$ and $\langle 0^e 0^{ph} | \frac{1}{2} (\lambda \cdot \Delta \mathbf{d})^2 | 0^e 0^{ph} \rangle$ corresponds to electronic and dipole self-energy contributions, respectively, calculated with the QED-Hartree-Fock method. The calculations employ cavity parameters $\lambda = 0.1$ a.u. and $\omega = 1.5$ eV. The Supplemental Table III indicate that the electronic energy contributes the most to the energy, whereas the changes in energy due to the cavity are mainly due to the dipole self-energy contributions. These changes in the dipole self-energy are due to the quantum vacuum fluctuations that are the main driving forces for the reactions inside a cavity. "

Comment 6:

Despite this, I find the study an interesting contribution to the treatment of light-matter coupling and it's potential effects on molecular reactivity interesting and publishable elsewhere (not in Nat. Commun.). .

We thank the reviewer for the very helpful and positive feedback and for finding our study interesting and we hope that this manuscript is now suitable for publication in *Nature Communications*.

Reviewer: 3

Comments:

In this theoretical study, it is shown that the reaction products of a couple of Diels-Alder reactions under electronic strong coupling depends on the orientation of the molecules relative to the cavity field. This is yet another demonstration of the potential of polaritonic chemistry and as such it will be of broad interest. The only troublesome points are the following:

We thank the reviewer for the thoughtful comments and we have modified the article to clarify the main issues raised by the reviewer. Please see our comments and corresponding modifications in the manuscript below.

Comment 1:

The proposed reactant molecules have electronic transitions only in the deep UV (ca. 5 eV for cyclopentadiene and even higher energies for the nitriles) not at 1.5 eV which seems to be the basis for the calculations (Fig. 3 left panel and discussion on pages 4 and 5). The authors should say more clearly what they are coupling and demonstrate that they are in the strong coupling regime.

We appreciate the reviewer's comment and for raising this important point. We agree that an additional clarification for the selection of the cavity frequency is necessary and the reviewer is correct that the excitation energies of the reaction constituents are higher than the employed frequency. Indeed, the calculated first excitation energy with the EOM-CCSD/cc-pVDZ method for both *endo* and *exo* reaction complex between CPD and AN is 5.8 eV. However, in this work the driving force for the observed changes of reaction barriers and energies is not due to formation of the excited state polaritonic states, but rather due to the quantum vacuum fluctuations caused mainly by the fluctuations of the dipole self-energy. Because the observed effect is not due to resonance of the cavity with respect to electronic excitations, our choice of cavity frequency was motivated by experimentally realizable values of the cavity frequency. Also as shown in Fig 3. and in Section 4 of the Supplemental Information the choice of cavity frequency has little effect on reactivity and no resonance is observed within this region. To indicate that, we have modified two sentences on Page 5 of the manuscript that now reads as:

"Note that selected values of the cavity frequency are well above any molecular vibrational levels and below electronic excitations [24, 25] and the selected range was motivated by the experimentally realizable values. As shown in Fig. 3, the content of the final product shows very little dependence on the cavity frequency and no resonance effect is observed, although the effect of the cavity is the greatest at low values and slowly decreases as the cavity frequency increases."

To additionally clarify that the observed change in reactivity is not due to formation of the excited state polaritons but rather quantum vacuum fluctuations, we have added a sentence to the Introduction on Page 2 of the main manuscript:

"The observed changes in the reactivity are due to the formation of a new light matter hybrid ground state via the quantum vacuum fluctuations."

Moreover, to support this claim we have calculated the individual energy contributions (electronic, dipole self-energy, and dipolar coupling) for the QED-CCSD method and we have added the following statement on Page 4 of the manuscript:

"To gain further understanding of the underlying mechanism, we have computed the individual contributions of the electronic, dipole self-energy, and dipolar coupling terms to the reaction barrier (see Section 6 of the Supplemental Information). When the light is polarized along the x molecular direction, both the electronic and dipole self-energy terms contribute significantly to the change in the reaction barrier, whereas in the case when the light is polarized in the y and z molecular directions, the change in the reaction barrier is mainly due to the dipole self-energy term."

as well as an additional section "Energy Contributions Breakdown" in the Supplemental Information with accompanying Supplemental Table III.

Lastly, as for the reviewer's comment for the demonstration that we are operating in the strong coupling regime, we refer to recent experiments in picocavity setups that have demonstrated the strong light-matter coupling strength > 0.1 a.u. [15, 16, 17, 18], thus achieving strong light-matter coupling in the single molecule limit. Alternatively, we could in principle calculate Rabi splitting for a given coupling strength, however, this would require EOM-QED-CCSD method that we have not implemented yet. However, in our previous work, we have used TD-QEDFT where we have shown that for the same value of strong light-matter coupling ($\lambda = 0.1$ a.u.) the ratio between Rabi splitting to the cavity frequency is 12% which is well within the strong light-matter coupling regime [19]. In order to address the reviewer's comment, we have added one sentence to Page 3 of the manuscript:

"Throughout this work, we work in the strong light-matter coupling regime by considering a $|\lambda| = 0.1$ a.u. [15, 16, 17, 18, 19]."

Comment 2:

Secondly, the tiny quantization volumes are not realistic for any practical condition. This should be discussed.

We thank the reviewer for making this valuable remark. In the present study, the employed light-matter coupling strength is set to 0.1 a.u. that corresponds to effective cavity volume of $V_{\text{eff}} = 0.19 \text{ nm}^3$ in the case of the single-mode cavity. However, in the case of the multi-mode cavities, the same effective light-matter coupling strength can be achieved with much larger cavity volumes via $\lambda_{\text{eff}}^2 = \sum_n \lambda_n^2$, where λ_n is the coupling strength of the n -th mode. Therefore, in the multimode cavities that were experimentally realized by Lev and co-workers [6], the light-matter coupling can be greatly enhanced by combining multiple cavity modes while keeping relatively

large volumes. Therefore, we are not bound to use the cavities with the small quantization volumes indicated by the reviewer due to available multimode cavities. In order to address the reviewer's concern, we have replaced a paragraph on Page 3 of the manuscript:

"Finally, we discuss in more detail the specifics of the cavity. The light-matter coupling strength in Eq. 1. is defined by $|\lambda_\alpha| = 1/\sqrt{\epsilon_0 V_\alpha^{\text{eff}}}$. Here, ϵ_0 is the permittivity of the vacuum and V_α^{eff} is the effective quantization volume [7] that depends on the nature of the cavity. In fact, recent advances in optical cavity design [8, 9, 10, 11] realize effective volumes as low as 0.15 nm^3 [12] which corresponds to strong light-matter coupling strength above 0.1 a.u and enable the possibility of controlling the molecular orientation inside the cavity [1]. Throughout this work, we work in the strong light-matter coupling regime by considering a $\lambda = 0.1$ a.u. that corresponds to $V_{\text{eff}} = 0.19 \text{ nm}^3$ [13]. This value of the coupling constant is within the experimental range [14]. Further details are provided in the Section 3 of the Supplemental Information."

with the following paragraph:

"Finally, we discuss in more detail the specifics of the cavity. The light-matter coupling strength in Eq. 1. is defined by $|\lambda_\alpha| = 1/\sqrt{\epsilon_0 V_\alpha^{\text{eff}}}$. Here, ϵ_0 is the permittivity of the vacuum and V_α^{eff} is the effective quantization volume [7] of a single mode cavity. Recent advances in the multi-mode cavity design [6, 8, 9, 10, 11] are able to realize the strong light-matter coupling strength above 0.1 a.u. and also enable the possibility of controlling the molecular orientation inside the cavity [1]. In the multi-mode cavities [6], the light-matter coupling strength can be greatly enhanced by increasing the number of cavity modes via $\lambda_{\text{eff}}^2 = \sum_n \lambda_n^2$, where λ_n is the coupling strength of the n -th cavity mode. Throughout this work, we work in the strong light-matter coupling regime by considering a $|\lambda| = 0.1$ a.u. [15, 16, 17, 18, 19]."

We thank the reviewer for the very helpful and positive feedback and hope we that this manuscript is now suitable for publication in *Nature Communications*.

References

- [1] Rohit Chikkaraddy, Bart de Nijs, Felix Benz, Steven J. Barrow, Oren A. Scherman, Edina Rosta, Angela Demetriadou, Peter Fox, Ortwin Hess, and Jeremy J. Baumberg. Single-molecule strong coupling at room temperature in plasmonic nanocavities. *Nature*, 535(7610):127–130, jun 2016.
- [2] Albert C Aragonès, Naomi L Haworth, Nadim Darwish, Simone Ciampi, Nathaniel J Bloomfield, Gordon G Wallace, Ismael Diez-Perez, and Michelle L Coote. Electrostatic catalysis of a diels–alder reaction. *Nature*, 531(7592):88–91, 2016.

- [3] Bretislav Friedrich. The exacting task of bringing molecules to attention. *Scientia*, 115(115):26–31, 2017.
- [4] Abhijit Sau, Kalaivanan Nagarajan, Bianca Patrahau, Lucas Lethuillier-Karl, Robrecht MA Vergauwe, Anoop Thomas, Joseph Moran, Cyriaque Genet, and Thomas W Ebbesen. Modifying woodward–hoffmann stereoselectivity under vibrational strong coupling. *Angewandte Chemie International Edition*, 60(11):5712–5717, 2021.
- [5] Vildan Guner, Kelli S Khuong, Andrew G Leach, Patrick S Lee, Michael D Bartberger, and KN Houk. A standard set of pericyclic reactions of hydrocarbons for the benchmarking of computational methods: The performance of ab initio, density functional, casscf, caspt2, and cbs-qb3 methods for the prediction of activation barriers, reaction energetics, and transition state geometries. *J. Phys. Chem. A*, 107(51):11445–11459, 2003.
- [6] Varun D Vaidya, Yudan Guo, Ronen M Kroeze, Kyle E Ballantine, Alicia J Kollár, Jonathan Keeling, and Benjamin L Lev. Tunable-range, photon-mediated atomic interactions in multi-mode cavity qed. *Phys. Rev. X*, 8(1):011002, 2018.
- [7] Javier Galego, Clàudia Climent, Francisco J. Garcia-Vidal, and Johannes Feist. Cavity casimir-polder forces and their effects in ground-state chemical reactivity. *Phys. Rev. X*, 9:021057, Jun 2019.
- [8] Fábio Barachati, Antonio Fieramosca, Soroush Hafezian, Jie Gu, Biswanath Chakraborty, Dario Ballarini, Ludvik Martinu, Vinod Menon, Daniele Sanvitto, and Stéphane Kéna-Cohen. Interacting polariton fluids in a monolayer of tungsten disulfide. *Nat. Nanotechnol.*, 13(10):906–909, 2018.
- [9] Weilu Gao, Xinwei Li, Motoaki Bamba, and Junichiro Kono. Continuous transition between weak and ultrastrong coupling through exceptional points in carbon nanotube microcavity exciton–polaritons. *Nat. Photon.*, 12(6):362–367, 2018.
- [10] Fuyang Tay, Ali Mojibpour, Shuang Liang, Andrey Baydin, Arash Ahmadvand, Nicolas Marquez Peraca, Hongjing Xu, Geoff C Gardner, Michael J Manfra, David Hagenmüller, et al. Landau polaritons in a full-dielectric three-dimensional photonic-crystal cavity. In *CLEO: Science and Innovations*, pages SW4G–6. Optica Publishing Group, 2022.
- [11] Felice Appugliese, Josefina Enkner, Gian Lorenzo Paravicini-Bagliani, Mattias Beck, Christian Reichl, Werner Wegscheider, Giacomo Scalari, Cristiano Ciuti, and Jérôme Faist. Breakdown of topological protection by cavity vacuum fields in the integer quantum hall effect. *Science*, 375(6584):1030–1034, 2022.
- [12] Tong Wu, Wei Yan, and Philippe Lalanne. Bright plasmons with cubic nanometer mode volumes through mode hybridization. *ACS Photonics*, 8(1):307–314, 2021.
- [13] Anton Frisk Kockum, Adam Miranowicz, Simone De Liberato, Salvatore Savasta, and Franco Nori. Ultrastrong coupling between light and matter. *Nat. Rev. Phys.*, 1(1):19–40, 2019.

- [14] Elad Eizner, Luis A. Martínez-Martínez, Joel Yuen-Zhou, and Stéphane Kéna-Cohen. Inverting singlet and triplet excited states using strong light-matter coupling. *Sci. Adv.*, 5(12):eaax4482, 2019.
- [15] Felix Benz, Mikolaj K. Schmidt, Alexander Dreismann, Rohit Chikkaraddy, Yao Zhang, Angela Demetriadou, Cloudy Carnegie, Hamid Ohadi, Bart de Nijs, Ruben Esteban, Javier Aizpurua, and Jeremy J. Baumberg. Single-molecule optomechanics in picocavities. *Science*, 354(6313):726–729, 2016.
- [16] Marie Richard-Lacroix and Volker Deckert. Direct molecular-level near-field plasmon and temperature assessment in a single plasmonic hotspot. *Light: Science & Applications*, 9(1), mar 2020.
- [17] Shuyi Liu, Adnan Hammud, Martin Wolf, and Takashi Kumagai. Atomic point contact raman spectroscopy of a si(111)-7 × 7 surface. *Nano Lett.*, 21(9):4057–4061, 2021.
- [18] Jack Griffiths, Bart de Nijs, Rohit Chikkaraddy, and Jeremy J. Baumberg. Locating single-atom optical picocavities using wavelength-multiplexed raman scattering. *ACS Photonics*, 8(10):2868–2875, 2021.
- [19] Fabijan Pavošević, Sharon Hammes-Schiffer, Angel Rubio, and Johannes Flick. Cavity-modulated proton transfer reactions. *J. Am. Chem. Soc.*, 144(11):4995–5002, 2022.
- [20] Lachlan P Lindoy, Arkajit Mandal, and David R Reichman. Quantum dynamics of vibrational polariton chemistry. *arXiv preprint arXiv:2210.05550*, 2022.
- [21] Dominik Sidler, Michael Ruggenthaler, Christian Schäfer, Enrico Ronca, and Angel Rubio. A perspective on ab initio modeling of polaritonic chemistry: The role of non-equilibrium effects and quantum collectivity. *J. Chem. Phys.*, 156(23):230901, 2022.
- [22] Christian Schäfer, Johannes Flick, Enrico Ronca, Prineha Narang, and Angel Rubio. Shining light on the microscopic resonant mechanism responsible for cavity-mediated chemical reactivity. *arXiv preprint arXiv:2104.12429*, 2021.
- [23] Christoph Riplinger and Frank Neese. An efficient and near linear scaling pair natural orbital based local coupled cluster method. *J. Chem. Phys.*, 138(3):034106, 2013.
- [24] Xiaoze Liu, Tal Galfsky, Zheng Sun, Fengnian Xia, Erh-chen Lin, Yi-Hsien Lee, Stéphane Kéna-Cohen, and Vinod M Menon. Strong light–matter coupling in two-dimensional atomic crystals. *Nat. Photon.*, 9(1):30–34, 2015.
- [25] Xinwei Li, Motoaki Bamba, Qi Zhang, Saeed Fallahi, Geoff C Gardner, Weilu Gao, Minhan Lou, Katsumasa Yoshioka, Michael J Manfra, and Junichiro Kono. Vacuum bloch–siebert shift in landau polaritons with ultra-high cooperativity. *Nat. Photon.*, 12(6):324–329, 2018.

REVIEWER COMMENTS

Reviewer #2 (Remarks to the Author):

I have read the replies of the authors to the comments of my colleagues and myself. Despite my appreciation for the work done by the authors, I am not fully convinced:

1) Mode volumes and feasibility of the proposal: the authors suggest to use a multimode optical cavity and exploit the coupling of the molecule to multiple modes of such a cavity to boost the effective light-matter interaction. In this scenario, if I understand it correctly, each mode would off-resonantly couple to the molecule well within the validity of the dipolar approximation of light-matter coupling, but collectively, the coupling would mimic the one of a cavity mode of a small mode volume ($\sim 0.1 \text{ nm}^3$).

Although this is possible (to some extent unavoidable as one always finds multiple modes in cavities), I am not convinced that the effective mode volume can be decreased while maintaining the assumption that the field is homogeneous-enough across the size of the molecule for the dipolar approximation to hold. I would be convinced if the authors present an actual solution of Maxwell's equations that satisfies the criteria outlined above (small mode volume $\ll 1 \text{ nm}^3$ while not breaking the dipole approximation). Isn't the effective mode volume directly linked with the field localization, even if achieved as a superposition of many modes? An example of such a scenario would be the well-known concept of plasmonic pseudomode, where a large number of higher-order plasmonic modes superimpose into a localized "effective" mode. In this case, the modes separately can have a larger mode volume, but collectively they behave as a mode of a much smaller volume. However, in real space, this "pseudomode" indeed features very strong localization of its electromagnetic field - potentially thus violating dipole selection rules. Moreover, it is important to take into account, that off-resonant coupling will strongly depend on the detuning of the cavity frequency from the energy of the relevant transitions in the molecule. The overall impact of light-matter coupling in this complicated multimode scenario is therefore not obvious to me - especially it is not obvious that a simple redefinition of the mode volume is sufficient.

Even if we find an optical (plasmonic) cavity with a single mode of very small mode volume, such as picocavities suggested by Baumberg et al., the molecule would likely not be able to take the full advantage of the mode volume comparable to its geometrical dimensions. That would only be possible for a point-like emitter placed into the position of maximal electric-field strength (usually on the surface of the plasmonic particle). That is, I find it geometrically hard - if not impossible - to imagine that such coupling strength can be achieved even in cavities of small effective volumes when realistic dimensions of molecules are considered (even if dipolar approximation is not invoked).

Also, in plasmonic cavities I would expect the electromagnetic (EM) field to be primarily of a longitudinal character, and therefore including transverse EM fields using the method of the authors may not be strictly necessary (the fields are already included in the Coulomb interaction present in (TD-)DFT and other standard methods).

I want to remark that other strategies to boost light-matter coupling have been proposed, one of them taking advantage of interactions mediated by surrounding molecules [S. Schütz et al. PRL, 124(11), 113602, 2020]. Maybe the authors could also try to consider such scenarios in their search for a realistic cavity.

To sum up, my main concern is that the values of light-matter coupling proposed by the authors may not be achievable in reality, but I could be convinced about the opposite. I would be convinced if the authors present an actual concrete solution of Maxwell's equations that satisfies the criteria outlined above (small mode volume $\ll 1 \text{ nm}^3$ while not breaking the

dipole approximation). Ideally, they should then demonstrate using their model that the effects discussed in the paper can still be achieved.

As a side remark, the alternative to couple multiple molecules to the cavity to effectively boost the light-matter coupling strength (collective coupling) would, in the case of a reaction having two reactants, be rather hard to imagine as one would have to prepare a large number of suitably oriented molecular pairs... This, in my view, renders the scenario somewhat artificial as well.

2) Fixing the position of the molecular pair (reactants) in the cavity:

The use of an external electric field suggested by the authors is certainly an option to partially control the orientation, but:

i - it is hard to imagine that all rotational and translational degrees of freedom of a pair of molecules could be controlled in this way

ii - the external electric field would certainly modify the electronic structure of the molecules with unforeseen effect on their chemical reactivity

Using the strategy of Baumberg et al. using molecular cages is appealing, but involves plasmonic cavities, proximity of metal surfaces, and the presence of the molecular cage itself. All this introduces effects well beyond the description introduced by the authors in the present manuscript. I suspect that these effects would play a much more important role than the off-resonant coupling with the EM modes of the cavity considered by the authors. Besides, the treatment of such plasmonic cavities is well within the scope of existing TD-DFT schemes where e.g. metal clusters can be used to model the effect of a plasmonic cavity in much more detail (and including different effects) than the method presented by the authors.

To summarize, I still believe that the work is publishable, but in a specialized journal (not Nature Communications), unless the authors present conclusive convincing arguments that their theoretical proposal meets realistic conditions (is experimentally verifiable in some way). Otherwise, I would say that the work is a nice theoretical exercise, but of not so broad interest.

Reviewer #3 (Remarks to the Author):

The authors have responded in a satisfactory manner to the referee comment and I recommend acceptance. Minor comment for the authors to think about: even if the change in chemical reactivity concerns the ground state, for the latter to gain polaritonic or photonic character, a transition dipole moment need to be coupled which it is not at 1.5 eV in the system considered.

Response to Reviewers: NCOMMS-22-37279A

The reviewer comments are shown in *italic* and our reply is in normal font.

Reviewer: 2

Comments:

I have read the replies of the authors to the comments of my colleagues and myself. Despite my appreciation for the work done by the authors, I am not fully convinced:

We thank the reviewer for the thoughtful comments and we have modified the article to clarify the main issues raised by the reviewer. Please see our comments and corresponding modifications made in the manuscript.

Comment 1:

Mode volumes and feasibility of the proposal: the authors suggest to use a multimode optical cavity and exploit the coupling of the molecule to multiple modes of such a cavity to boost the effective light-matter interaction. In this scenario, if I understand it correctly, each mode would off-resonantly couple to the molecule well within the validity of the dipolar approximation of light-matter coupling, but collectively, the coupling would mimic the one of a cavity mode of a small mode volume (0.1 nm^3).

Although this is possible (to some extent unavoidable as one always finds multiple modes in cavities), I am not convinced that the effective mode volume can be decreased while maintaining the assumption that the field is homogeneous-enough across the size of the molecule for the dipolar approximation to hold. I would be convinced if the authors present an actual solution of Maxwell's equations that satisfies the criteria outlined above (small mode volume $\ll 1 \text{ nm}^3$ while not breaking the dipole approximation). Isn't the effective mode volume directly linked with the field localization, even if achieved as a superposition of many modes? An example of such a scenario would be the well-known concept of plasmonic pseudo-mode, where a large number of higher-order plasmonic modes superimpose into a localized "effective" mode. In this case, the modes separately can have a larger mode volume, but collectively they behave as a mode of a much smaller volume. However, in real space, this "pseudo-mode" indeed features very strong localization of its electromagnetic field - potentially thus violating dipole selection rules. Moreover, it is important to take into account, that off-resonant coupling will strongly depend on the detuning of the cavity frequency from the energy of the relevant transitions in the molecule. The overall impact of light-matter coupling in this complicated multimode scenario is therefore not obvious to me - especially it is not obvious that a simple redefinition of the mode volume is sufficient.

Even if we find an optical (plasmonic) cavity with a single mode of very small mode volume, such as picocavities suggested by Baumberg et al., the molecule would likely not be able to take the full advantage of the mode volume comparable to its geometrical dimensions. That would only be possible for a point-like emitter placed into the position of maximal electric-field strength (usually on the surface of the plasmonic particle). That is, I find it geometrically hard - if not impossible - to

imagine that such coupling strength can be achieved even in cavities of small effective volumes when realistic dimensions of molecules are considered (even if dipolar approximation is not invoked).

Also, in plasmonic cavities I would expect the electromagnetic (EM) field to be primarily of a longitudinal character, and therefore including transverse EM fields using the method of the authors may not be strictly necessary (the fields are already included in the Coulomb interaction present in (TD-)DFT and other standard methods).

I want to remark that other strategies to boost light-matter coupling have been proposed, one of them taking advantage of interactions mediated by surrounding molecules [S. Schütz et al. PRL, 124(11), 113602, 2020]. Maybe the authors could also try to consider such scenarios in their search for a realistic cavity.

To sum up, my main concern is that the values of light-matter coupling proposed by the authors may not be achievable in reality, but I could be convinced about the opposite. I would be convinced if the authors present an actual concrete solution of Maxwell's equations that satisfies the criteria outlined above (small mode volume $\ll 1\text{nm}^3$ while not breaking the dipole approximation). Ideally, they should then demonstrate using their model that the effects discussed in the paper can still be achieved.

As a side remark, the alternative to couple multiple molecules to the cavity to effectively boost the light-matter coupling strength (collective coupling) would, in the case of a reaction having two reactants, be rather hard to imagine as one would have to prepare a large number of suitably oriented molecular pairs... This, in my view, renders the scenario somewhat artificial as well.

We thank the reviewer for the relevant comments. We would like to emphasise that the use of multimode cavity is not the only way for increasing the light-matter coupling. As the reviewer pointed out, for which comment we are really grateful, there are other interesting ways for boosting this effect [S. Schütz et al. PRL, 124(11), 113602, 2020]. We would also like to stress out that for the dipole approximation to hold, it is necessary to locate the molecular system in the center of the cavity where the mode-field can be approximated by a constant which can be achieved with the confocal cavities [1]. This assumes that the cavity is much larger than the molecular system of study. Therefore, for reaching the desired coupling strengths in the experiment we are not bound to extremely small cavity volumes. Effects beyond the dipole approximation are currently not included in our approach, but it will be interesting to study in the future how much they are contributing in this regime.

To address the reviewer's comment, we have included two sentences on Page 3 of the manuscript:

"We note that for the dipole approximation to hold, it is necessary to locate the molecular system in the center of the cavity where the mode-field can be approximated by a constant as realized with the confocal cavities [1], allowing that the cavity volume is much larger than the molecular system of interest."

and:

"Alternative way for boosting the light matter coupling can be achieved by taking advantage of interactions mediated by ensemble of nearby emitters [2]."

Comment 2:

Fixing the position of the molecular pair (reactants) in the cavity: The use of an external electric field suggested by the authors is certainly an option to partially control the orientation, but:

(i) - it is hard to imagine that all rotational and transnational degrees of freedom of a pair of molecules could be controlled in this way

(ii) - the external electric field would certainly modify the electronic structure of the molecules with unforeseen effect on their chemical reactivity.

We thank the reviewer for this comment. As demonstrated experimentally, a polar molecule can be aligned with the direction of the external electric field through the interaction of its dipole moment and polarizability with the applied field [3, 4, 5, 6, 7, 8].

Moreover, the application of the external electric field would cause a well defined changes to chemical reactivity in the Diels-Alder reactions as demonstrated both theoretically [9, 10] and experimentally [11, 12]. In particular, application of the electric field along the reaction axis will lower the energy of transition state by changing the bond ionicity and by mixing of charge-transfer states, whereas application of the electric field oriented perpendicular to the reaction axis does not affect the reactivity [12]. It was demonstrated experimentally that in the case of a two-step cascade reaction of the Diels-Alder reaction followed by an aromatization reaction, the reaction rate is increased by 10 times when the electric field is aligned with the reaction axis, whereas the reaction rate remains unchanged when the reaction axis is orthogonal to the electric field [12]. In our case for the cavity modulated Diels-Alder reactions, we found that the reaction rate decreases by ~10,000 times if the cavity mode is polarized along the reaction axis, whereas for the case when the cavity mode is polarized perpendicular to the reaction axis the rate is increased by ~100 times. Therefore, presumably these two processes, application of the external field and strong light-matter interaction are not competing processes. To address the reviewer's comment, we have added two sentences on Page 7 of the manuscript:

"As experimentally confirmed, application of external static fields would increase the reaction rate only if the electric field is aligned with the reaction axis, otherwise leaving the reaction rate unchanged [12]. Because of that, observed *endo/exo* selectivity due to strong light-matter would not be affected by the presence of the external electric fields."

Comment 3:

Using the strategy of Baumberg et al. using molecular cages is appealing, but involves plasmonic cavities, proximity of metal surfaces, and the presence of the molecular cage itself. All this introduces effects well beyond the description introduced by the authors in the present manuscript. I suspect that these effects would play a much more important role than the off-resonant coupling with the EM modes of the cavity considered by the authors. Besides, the treatment of such plasmonic cavities is well within the scope of existing TD-DFT schemes where e.g. metal clusters can be used to model the effect of a plasmonic cavity in much more detail (and including different effects) than the method presented by the authors.

We thank the reviewer for this comment. The interaction between localized surface plasmons and molecules is the subject of the recent article by Koch and his co-workers [13], however we do not consider the effects due to plasmon-molecule interaction which is beyond the scope of the current study.

To summarize, I still believe that the work is publishable, but in a specialized journal (not Nature Communications), unless the authors present conclusive convincing arguments that their theoretical proposal meets realistic conditions (is experimentally verifiable in some way). Otherwise, I would say that the work is a nice theoretical exercise, but of not so broad interest.

We thank the reviewer for the very helpful feedback and we hope that the manuscript with the additional information is now suitable for publication in *Nature Communications*.

Reviewer: 3

Comments:

The authors have responded in a satisfactory manner to the referee comment and I recommend acceptance. Minor comment for the authors to think about: even if the change in chemical reactivity concerns the ground state, for the latter to gain polaritonic or photonic character, a transition dipole moment need to be coupled which it is not at 1.5 eV in the system considered.

We thank the reviewer to stress that our manuscript is now suitable for publication in *Nature Communications*.

References

- [1] Varun D Vaidya, Yudan Guo, Ronen M Kroeze, Kyle E Ballantine, Alicia J Kollár, Jonathan Keeling, and Benjamin L Lev. Tunable-range, photon-mediated atomic interactions in multi-mode cavity qed. *Phys. Rev. X*, 8(1):011002, 2018.
- [2] Stefan Schütz, Johannes Schachenmayer, David Hagenmüller, Gavin K Brennen, Thomas Volz, Vahid Sandoghdar, Thomas W Ebbesen, Claudiu Genes, and Guido Pupillo. Ensemble-induced strong light-matter coupling of a single quantum emitter. *Phys. Rev. Lett.*, 124(11):113602, 2020.
- [3] KH Kramer and RB Bernstein. Focusing and orientation of symmetric-top molecules with the electric six-pole field. *J. Chem. Phys.*, 42(2):767–770, 1965.

- [4] Bretislav Friedrich and Dudley R Herschbach. Spatial orientation of molecules in strong electric fields and evidence for pendular states. *Nature*, 353(6343):412–414, 1991.
- [5] Kouichi Hayashi, Shin'ichi Kawato, Yasuhiro Fujii, Toshihisa Horiuchi, and Kazumi Matsushige. Effect of applied electric field on the molecular orientation of epitaxially grown organic films. *Appl. Phys. Lett.*, 70(11):1384–1386, 1997.
- [6] Toby D Hain, Robert M Moision, and Thomas J Curtiss. Hexapole state-selection and orientation of asymmetric top molecules: CH_2F_2 . *J. Chem. Phys.*, 111(15):6797–6806, 1999.
- [7] Lewis J Martin, Behnam Akhavan, and Marcela MM Bilek. Electric fields control the orientation of peptides irreversibly immobilized on radical-functionalized surfaces. *Nat. Chem.*, 9(1):357, 2018.
- [8] Adam Jaroš, Esmail Farajpour Bonab, Michal Straka, and Cina Foroutan-Nejad. Fullerene-based switching molecular diodes controlled by oriented external electric fields. *J. Am. Chem. Soc.*, 141(50):19644–19654, 2019.
- [9] Rinat Meir, Hui Chen, Wenzhen Lai, and Sason Shaik. Oriented electric fields accelerate diels–alder reactions and control the endo/exo selectivity. *ChemPhysChem*, 11(1):301–310, 2010.
- [10] Sason Shaik, Debasish Mandal, and Rajeev Ramanan. Oriented electric fields as future smart reagents in chemistry. *Nat. Chem.*, 8(12):1091–1098, 2016.
- [11] Albert C Aragonés, Naomi L Haworth, Nadim Darwish, Simone Ciampi, Nathaniel J Bloomfield, Gordon G Wallace, Ismael Diez-Perez, and Michelle L Coote. Electrostatic catalysis of a diels–alder reaction. *Nature*, 531(7592):88–91, 2016.
- [12] Xiaoyan Huang, Chun Tang, Jieqiong Li, Li-Chuan Chen, Jueting Zheng, Pei Zhang, Jiabo Le, Ruihao Li, Xiaohui Li, Junyang Liu, et al. Electric field–induced selective catalysis of single-molecule reaction. *Sci. Adv.*, 5(6):eaaw3072, 2019.
- [13] Jacopo Fregoni, Tor S Haugland, Silvio Pipolo, Tommaso Giovannini, Henrik Koch, and Stefano Corni. Strong coupling between localized surface plasmons and molecules by coupled cluster theory. *Nano Lett.*, 21(15):6664–6670, 2021.

REVIEWERS' COMMENTS

Reviewer #2 (Remarks to the Author):

I have read the response of the authors, and I am still lacking more quantitative discussion regarding my concerns about the mode volume and appropriateness of the dipole approximation. I would still recommend them to actually perform the calculation estimating the effective light-matter coupling (mode volume) and the spatial variations of the EM field associated with it, regardless of the type of cavity that the authors propose (including the confocal cavity). Nonetheless, this is the responsibility of the authors, and I am therefore not going to insist on further calculations/modifications.

I appreciate the discussion of effects of the electric field on the molecular reactivity.

In conclusion, despite my remaining doubts, I believe that the paper can be published, if the editor is satisfied with its current form.

Response to Reviewers: NCOMMS-22-37279B

The reviewer comments are shown in *italic* and our reply is in normal font.

Reviewer: 2

We thank the reviewer for the thoughtful comments and we have modified the article to clarify the main issues raised by the reviewer. Please see our comments and corresponding modifications made in the manuscript.

Comment 1:

I have read the response of the authors, and I am still lacking more quantitative discussion regarding my concerns about the mode volume and appropriateness of the dipole approximation. I would still recommend them to actually perform the calculation estimating the effective light-matter coupling (mode volume) and the spatial variations of the EM field associated with it, regardless of the type of cavity that the authors propose (including the confocal cavity). Nonetheless, this is the responsibility of the authors, and I am therefore not going to insist on further calculations/modifications.

We thank the reviewer for this comment. To address the reviewer's comment and to avoid a confusion introduced by defining the light-matter coupling strength via the cavity volume, we have rewritten the subsection that discusses the specifics of the cavity. This discussion is now added in the "Methods" section under subsection "Specifics of the cavity". The major point of this discussion is that we don't necessarily need to use picocavities to achieve a large value of the light-matter coupling constant (~ 0.1 a.u.) that we are considering in this work. Other cavity setups with much larger volume are able to achieve such values experimentally. Next, we discuss that such high values can be achieved using a number of techniques, such as collectivity, multimode approach as well as others. Finally, we acknowledge that the dipole approximation would be insufficient for quantitatively accurate description of the reactions in nanocavities and picocavities, however in other cavity setups the dipole approximation is even quantitatively accurate.

The previous discussion has now been replaced by the following paragraph:

"Throughout this work, we work in the strong light-matter coupling regime by considering a $|\lambda| = 0.1$ a.u. [27, 33, 54-56]. Recent advances in the cavity design are able to realize the strong light-matter coupling to within this value such as achieved in case of the Landau polaritons [29], distributed Bragg reflector cavities [28], and in plasmonic nanocavities [43]. Boosting the light-matter coupling can be achieved by exploiting the collectivity effect [17, 25], by taking advantage of interactions mediated by ensemble of nearby emitters [57], by the multimode cavities [58], or by employing the picocavities [59]. We note that for description of the strong light-matter effects in the nanocavities and picocavities, one must go beyond the dipole approximation. The contributions beyond the dipole approximation may be relevant for obtaining the quantitatively accurate results, however the qualitative results that we show in here will not be modified. Additionally, for other cavity setups the dipole term would be the dominant one even in quantitative terms."

Comment 2:

I appreciate the discussion of effects of the electric field on the molecular reactivity.

We thank the reviewer for this comment and we are glad that adding this discussion has additionally improved the quality of our manuscript.

In conclusion, despite my remaining doubts, I believe that the paper can be published, if the editor is satisfied with its current form.

We thank the reviewer for the very helpful feedback and we hope that the manuscript with the additional information and changes is now suitable for publication in *Nature Communications*.